# Coral bacterial community structure responds to environmental change in a host-specific manner

Maren Ziegler [1,2,8], Carsten G. B. Grupstra[1,3,8], Marcelle M. Barreto[1], Martin Eaton[4], Jaafar BaOmar[5,6], Khalid Zubier[5], Abdulmohsin Al-Sofyani[5], Adnan J. Turki[5], Rupert Ormond [4,5] & Christian R. Voolstra [1,7]

The global decline of coral reefs heightens the need to understand how corals respond to changing environmental conditions. Corals are metaorganisms, so-called holobionts, and restructuring of the associated bacterial community has been suggested as a means of holobiont adaptation. However, the potential for restructuring of bacterial communities across coral species in different environments has not been systematically investigated. Here we show that bacterial community structure responds in a coral host-specific manner upon cross-transplantation between reef sites with differing levels of anthropogenic impact. The coral *Acropora hemprichii* harbors a highly flexible microbiome that differs between each level of anthropogenic impact to which the corals had been transplanted. In contrast, the microbiome of the coral *Pocillopora verrucosa* remains remarkably stable. Interestingly, upon cross-transplantation to unaffected sites, we find that microbiomes become indistinguishable from back-transplanted controls, suggesting the ability of microbiomes to recover. It remains unclear whether differences to associate with bacteria flexibly reflects different holobiont adaptation mechanisms to respond to environmental change.

[1] Red Sea Research Center, Division of Biological and Environmental Science and Engineering (BESE), 4700 King Abdullah University of Science and Technology (KAUST), Thuwal 23955, Saudi Arabia. [2] Department of Animal Ecology & Systematics, Justus Liebig University, Heinrich-Buff-Ring 26–32 IFZ, 35392 Giessen, Germany. [3] BioSciences Department, Rice University, 6100 Main Street, Houston, TX 77005, USA. [4] Centre for Marine Biology and Biodiversity, Institute for Earth and Life Sciences, Heriot-Watt University, Riccarton, Edinburgh EH14 4AS, UK. [5] Faculty of Marine Science, King Abdulaziz University, PO Box 80207Jeddah 21589, Saudi Arabia. [6] Department of Marine Biology, Faculty of Environmental Science and Marine Biology, Hadhramout University, Al-Mukalla, Republic of Yemen. [7] Department of Biology, University of Konstanz, 78457 Konstanz, Germany. [8]These authors contributed equally: Maren Ziegler, Carsten G. B. Grupstra. Correspondence and requests for materials should be addressed to M.Z. (email: maren.ziegler@bio.uni-giessen.de) or to C.R.V. (email: christian.voolstra@uni-konstanz.de)

Scleractinian corals live in close association with endosymbiotic dinoflagellates of the family Symbiodiniaceae and a diverse community of bacteria (among other microorganisms), collectively referred to as the microbiome[1]. The community of the coral host and its associated microbiome comprises a metaorganism and is referred to as the coral holobiont[2]. While most corals depend on Symbiodiniaceae to meet their energetic demands by the transfer of photosynthetically fixed carbon[3,4], bacterial microbiome members fulfill a range of other functions including nitrogen fixation, sulfur cycling, and protection against pathogenic bacteria[5–9]. Accordingly, restructuring of the microbiome is proposed to contribute to coral holobiont plasticity and adaptation[1,10–12].

Recent studies have found differences in the degree to which coral microbiomes vary over environmental gradients or experimental treatments[13–15]. For example, microbiomes of *Ctenactis echinata* varied between different reef habitats to the degree that abundance of coral host species was associated with the presence/absence of specific bacteria[16]. Further, microbial diversity was shown to increase with depth in several coral species, possibly allowing corals to access a broader range of food sources[17]. In addition, some studies have found seasonal fluctuations in coral-associated microbiomes[16,18,19] and tide-related shifts on much shorter time scales[20], while other corals maintain temporally stable microbiomes[21]. Unidirectional transplantation experiments of the coral species *Acropora muricata*[22] and *Porites cylindrica*[23] from a pristine site to impacted or modified sites further illustrate that microbiomes change under adverse environmental conditions. In contrast, a transplantation experiment between different thermal habitats showed that microbiomes of heat tolerant *Acropora hyacinthus* can be acquired by heat sensitive corals upon environmental transplantation over the course of 17 months[11]. Notably, these corals exhibited increased thermo-tolerance in a subsequent heat stress experiment, harboring a more robust and stable microbiome.

At present, it is unclear whether the potential for microbiome restructuring is a conserved trait across coral species or whether species-specific differences exist. For instance, the coral *Pocillopora verrucosa* shows a globally conserved association with its main bacterial symbiont *Endozoicomonas*[14] that remains unchanged even under conditions of bleaching and mortality[24]. *P. verrucosa* further was shown to maintain a stable Symbiodiniaceae community during a cross-transplantation experiment over depth[25] and between seasons and reefs, while *Porites lutea* sampled under the same conditions had a highly flexible Symbiodiniaceae community[26]. Therefore, it appears that the ability of corals to associate with distinct microbial associates may depend on location, environmental setting, and coral host species.

To assess potential differences in flexibility of bacterial association across coral species, we conducted a long-term large-scale reciprocal transplantation experiment using the coral species *Acropora hemprichii* and *P. verrucosa*. Based on previous studies, these coral genera were suspected to differ in the flexibility of their association with different microbial communities across environmental gradients[14,24,27,28]. The use of a reciprocal transplant design between reef sites subjected to different levels of anthropogenic impact allowed us to assess whether environmental differences align with distinct bacterial communities and whether the ability to adapt bacterial community composition differs between coral species. As coral microbiomes have previously been shown to recover from stress events, such as bleaching[29] or disease[30], we were interested to elucidate whether coral microbiomes can recover from chronic pollution, i.e. return to a state that resembles conspecific microbiomes at unaffected sites upon transplantation of coral fragments from affected to pristine sites.

Our study shows that the degree of bacterial community restructuring upon transplantation to reef sites with different levels of anthropogenic impact differs between host species. *A. hemprichii* harbors a highly flexible bacterial community that is characterized by many differentially abundant bacterial taxa between impacts. In contrast, the bacterial community of *P. verrucosa* remains remarkably stable between impacts. In addition, microbial communities recovered to their original states upon cross-transplantation to unaffected sites. Thus, distinct degrees of host-associated bacterial community restructuring exist, but their role in holobiont adaptation to environmental change is currently unknown.

## Results

**Sample and sequencing overview.** After 21 months of reciprocal transplantations between 5 reef sites representing different levels of anthropogenic impact in the vicinity of Jeddah, Saudi Arabia, 135 coral fragments and 20 seawater samples were collected and their bacterial communities investigated (Fig. 1). MiSeq sequencing of bacterial 16S rRNA gene amplicons resulted in 10,857,521 sequences from 131 samples and 2 negative controls (Supplementary Data 1). After quality control and removal of unwanted sequences, 2,509,787 sequences with a mean length of 291 bp were retained (31,920 distinct sequences). *A. hemprichii* samples contained an average of 16,267 sequences (range: 189–45,628), *P. verrucosa* samples 26,930 sequences (range 5–67,805), and seawater samples 4,993 sequences (range 3,101–9,801) (Supplementary Data 1). After subsampling to 3,101 sequences per sample, we retained 48 samples of *P. verrucosa* and 54 samples of *A. hemprichii* (Table 1). Clustering to 97% similarity yielded 7,032 distinct OTUs (Source Data, Supplementary Data 2). The mean number of OTUs per sample was 237 (range 58–466) for *A. hemprichii*, 159 (range 45–427) for *P. verrucosa*, and 174 (range 129–249) for seawater (Source Data).

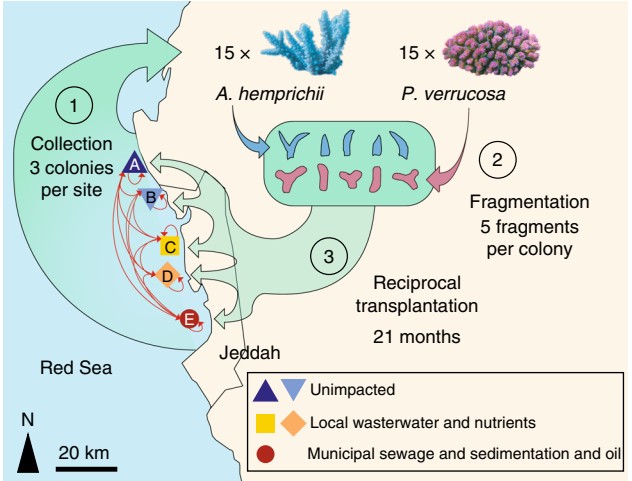

**Fig. 1** Design of transplantation experiment. Back-transplantation and cross-transplantation of coral fragments from *Acropora hemprichii* and *Pocillopora verrucosa* across sites of differing environmental impact close to the Saudi Arabian city of Jeddah in the Red Sea to test for microbiome flexibility. Three coral colonies of each species at each site were collected and fragmented into five fragments each per colony to allow for reciprocal transplantation across all sites. GPS locations for site A: N 21°52′22.83″ E 38°58′01.61″, site B: N 21°47′08.23″, E 39°02′28.56″, site C: N 21°36′54.53″, E 39°06′17.92″, site D: N 21°34′34.21″, E 39°06′27.27″, site E: N 21°26′21.41″, E 39°06′28.49″

**Table 1 Overview of transplanted coral fragments of *A. hemprichii* and *P. verrucosa***

| Destination impact (site) | Unimpacted (A) | Unimpacted (B) | Local wastewater and nutrients (C) | Local wastewater and nutrients (D) | Municipal sewage and sedimentation and oil (E) |
|---|---|---|---|---|---|
| Origin impact (site): | *A. hemprichii* | | | | |
| Unimpacted (A) | 3 | 0 | 1 | 2 | 1 |
| Unimpacted (B) | 3 | 3 | 1 | 2 | 3 |
| Local wastewater and nutrients (C) | 2 | 3 | 3 | 0 | 3 |
| Local wastewater and nutrients (D) | 3 | 3 | 1 | 2 | 3 |
| Municipal sewage and sedimentation and oil (E) | 3 | 3 | 1 | 2 | 3 |
| Samples per destination site | 14 | 12 | 7 | 8 | 13 |
| Samples per destination impact | 26 | | 15 | | 13 |
| Samples per origin site | 7 | 12 | 11 | 12 | 12 |
| Samples per origin impact | 19 | | 23 | | 12 |
| | | | | | |
| Origin impact (site): | *P. verrucosa* | | | | |
| Unimpacted (A) | 3 | 2 | 0 | 1 | 3 |
| Unimpacted (B) | 3 | 2 | 1 | 2 | 3 |
| Local wastewater and nutrients (C) | 3 | 3 | 2 | 1 | 2 |
| Local wastewater and nutrients (D) | 1 | 2 | 1 | 1 | 2 |
| Municipal sewage and sedimentation and oil (E) | 3 | 2 | 1 | 1 | 3 |
| Samples per destination site | 13 | 11 | 5 | 6 | 13 |
| Samples per destination impact | 24 | | 11 | | 13 |
| Samples per origin site | 9 | 11 | 11 | 7 | 10 |
| Samples per origin impact | 20 | | 18 | | 10 |

Shown are numbers of analyzed coral fragments after 21 months of reciprocal transplantation between 5 sites exposed to different levels of anthropogenic impact in the central Red Sea. Paired sites correspond to distinct impacts: A/B = unimpacted, C/D = Local wastewater and nutrients, E/lost site F = Municipal sewage and sedimentation and oil

**Microbial communities in water and coral samples are distinct.** Bacterial communities in seawater samples were distinct between individual sites, with the exception of sites B and E (PERMANOVA, $F = 2.2$, $P_{MC} = 0.001$; Supplementary Fig. 1). Seawater samples were dominated by the bacterial families Pelagibacteraceae (38–54%) and OCS 155 (14–31%) (Supplementary Fig. 2). Other abundant families included Flavobacteriaceae (4–13%) and Halomonadaceae (4–12%). Five additional taxa that were present at abundances >1% included Rhodobacteraceae (2–6%), Alteromonadaceae (0–10%), Rhodospirillaceae (1–2%), unclassified Alphaproteobacteria (1–6%), and unclassified Bacteria (0–3%). Microbial communities of the seawater were distinct from those of the two coral species (Supplementary Fig. 3). Of the 1,105 OTUs identified in water samples, only 172 (<16%) were shared with at least one of the coral species. In order to focus on differences between coral microbiomes, seawater samples were excluded from subsequent analyses.

**Coral microbiomes differ in their flexibility between sites.** A total of 4,704 bacterial OTUs were identified for *A. hemprichii* and 3,023 OTUs for *P. verrucosa*. Of these, 1,628 OTUs were shared between the two coral species. Bacterial assemblages were highly species-specific and significantly different in their multivariate dispersion between coral species (ANOVA, $F = 16.01$, $p < 0.001$; Fig. 2a). Overall, the bacterial communities of *A. hemprichii* samples were more variable than those of *P. verrucosa*, as evidenced by significantly higher distances to centroids (*A. hemprichii* mean = 0.55, SD = 0.05; *P. verrucosa* mean = 0.51, SD = 0.06, Supplementary Fig. 4, Source Data). After 21 months of reciprocal transplantations between 5 reef sites, the bacterial community of *A. hemprichii* differed significantly between all individual sites (Fig. 2b; Table 2). By comparison, differences in the bacterial community of *P. verrucosa* between sites were far less pronounced, with only fragments transplanted to the most highly impacted site (municipal sewage and sedimentation and oil —site E) being significantly different from coral fragments at the unimpacted sites A and B (Fig. 2c). The latter were also

significantly different from each other (Table 2). The differential pattern of bacterial microbiome restructuring between coral species was also evident when sites were pooled by impacts (Supplementary Data 3). Notably, analyzing the dataset using a 99% OTU similarity cutoff to account for putative differences at a higher phylogenetic resolution confirmed observed patterns across sites and impacts (pooled sites) (Supplementary Data 3–4). Similarly, excluding Endozoicomonadaceae from the dataset to rule out that patterns were largely driven by dominant association with *Endozoicomonas* also reproduced that patterns are different between both coral species when considering bacterial microbiome composition across sites and impacts (pooled sites) (Supplementary Data 3, 5). The site of origin, where the coral fragments were originally collected, had no significant effect on coral microbiome structuring (Table 2, Supplementary Figs. 5–8).

**Differentially abundant taxa align with microbiome flexibility.** In *A. hemprichii* pronounced microbiome restructuring between different impact levels was aligned with large shifts in abundance of 60 specific bacterial lineages, whereas *P. verrucosa* was associated with a much more stable and less variant microbial community, only 5 significantly different taxa being recorded between impacts (Fig. 3). *A. hemprichii* colony fragments transplanted to unimpacted sites had higher abundances of the bacterial family Endozoicomonadaceae (31% compared to 1–5%, Fig. 4a), as highlighted by OTU00003 (99% identical to *Endozoicomonas acroporae* GenBank accession no. NR_158127.1), which was significantly more abundant at unimpacted sites (LEfSe, LDA = 5.2, $p < 0.05$; Fig. 3a). Further bacterial taxa that occurred at higher abundance at unimpacted sites included Alteromonadales (23% compared to 11–26%) and Simkaniaceae (2% compared to 0–1%) (Fig. 4a). In contrast, several bacterial families showed impact-specific increases in their relative abundance. For instance, Caulobacteraceae and Comamonadaceae (both 2–6%) with 1 and 4 OTUs, respectively, including their representatives *Caulobacter* sp. (OTU00007, 99% identical

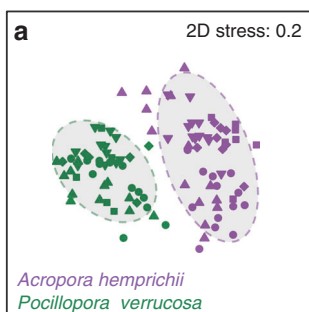
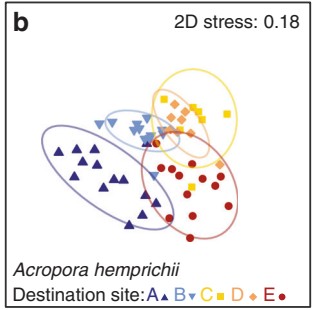
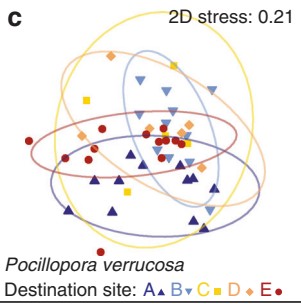

**Fig. 2** Bacterial community structure and relative dispersion of the coral species *A. hemprichii* and *P. verrucosa*. All plots based on non-metric multidimensional scaling (nMDS) of Bray-Curtis distances of bacterial community compositions associated with coral samples (after 21 months of reciprocal transplantation). **a** corals *A. hemprichii* and *P. verrucosa*. **b** only coral *A. hemprichii* with sites indicated. **c** only coral *P. verrucosa* with sites indicated. A/B unimpacted sites; C/D local wastewater and nutrients sites; E municipal sewage and sedimentation and oil site. Ellipses denote 90% confidence intervals. Source data are provided as a Source Data file

**Table 2 Statistical overview over bacterial microbiome differences across sites**

| Two-way PERMANOVA test statistics | | | Pairwise comparisons by destination site | | | | | |
|---|---|---|---|---|---|---|---|---|
| Factor | Model-F | $P_{MC}$ | | A | B | C | D | E |
| *Species: A. hemprichii* | | | A | – | 3.4 | 3.62 | 3.7 | 3.28 |
| Destination site | 2.86 | **<0.001** | B | **0.01** | – | 2.55 | 1.1 | 3.33 |
| Origin site | 1.1 | 0.14 | C | **0.01** | **0.01** | – | 1.36 | 2.02 |
| Destination × Origin site | 0.98 | 0.6 | D | **0.01** | **0.01** | **0.01** | – | 2.17 |
| | | | E | **0.01** | **0.01** | **0.01** | **0.01** | – |
| *Species: P. verrucosa* | | | A | – | 2.48 | 1.76 | 1.9 | 2.14 |
| Destination site | 1.82 | **<0.001** | B | **0.01** | – | 1.54 | 1.4 | 1.92 |
| Origin site | 1.1 | 0.18 | C | 0.08 | 0.11 | – | 1.19 | 1.22 |
| Destination × Origin site | 1.07 | 0.17 | D | 0.17 | 0.26 | 1.00 | – | 1.26 |
| | | | E | **0.04** | **0.03** | 1.00 | 1.00 | – |

Significant comparisons are printed in bold
$P_{MC}$: Monte-Carlo permuted *p*-value
Pairwise comparisons: lower triangle *p*-values, upper triangle Model-*F* values
Tests were conducted separately for the corals *Acropora hemprichii* and *Pocillopora verrucosa* after 21 months of reciprocal transplantation

to *Caulobacter* sp. Genbank AB470462.1, LEfSe, LDA = 4.3, *p* < 0.05) and *Pelomonas* sp. (OTU00026, 100% identical to *Pelomonas sp.* GenBank MF400787.1, LEfSe, LDA = 4.0, *p* < 0.05) were significantly more abundant at local wastewater and nutrients sites (Figs. 3a, 4a). Furthermore, Flavobacteriaceae (2–4%, 3 OTUs, incl. OTU0017, 100% identical to *Flavobacterium* sp. GenBank MK216896.1, LEfSe, LDA = 3.7, *p* < 0.05) and Rhodobacteraceae (5–15%, 3 OTUs, incl. OTU0178, 100% identical to *Roseovarius* sp. GenBank NR_108232.1, LEfSe, LDA = 3.1, *p* < 0.05) were significantly more abundant at the municipal sewage and sedimentation and oil site. In contrast, the stable microbiome of *P. verrucosa* was dominated by the bacterial family Endozoicomonadaceae at all sites (56–71%, Fig. 4b). The only dominant bacterial family that noticeably changed in abundance was the Simkaniaceae with a decrease from 17% at unimpacted sites to 5 and 0% at impacted sites (OTU0005, 100% identical to *Simkania* sp. GenBank NR_074932.1, LEfSe, LDA = 5.0, *p* < 0.05, Figs. 3b, 4b).

**Microbial indicator taxa.** In line with substantial differences in the bacterial association of *A. hemprichii* across sites, and a rather stable microbiome of *P. verrucosa*, we identified 62 bacterial indicator taxa (indicspecies analysis) for impacts of destination in *A. hemprichii*, nearly nine times as many as for *P. verrucosa* (Supplementary Data 6). When considering bacterial indicator taxa for the impact of origin, only 5 taxa were identified for *A. hemprichii* and none for *P. verrucosa*.

**Coral core microbiome.** To determine a putative core microbiome, all bacterial OTUs that were found at all sites in ≥75% of samples per site for a given coral host were considered. This led us to identify 7 bacterial taxa consistently associated with *A. hemprichii*. Among these were the second most abundant OTU0002 (*Melitea* sp.) as well as OTU0007 (*Caulobacter* sp.) and OTU0017 (*Flavobacterium* sp.), which were both present in all samples of *A. hemprichii*. For *P. verrucosa*, we identified 6 consistently associated taxa, of which the most abundant OTU0001 (Endozoicomonadaceae) was present in all samples of *P. verrucosa*. We further found a second Endozoicomonadaceae OTU consistently associated with *P. verrucosa*, but at much lower abundances. Three taxa were shared between the core microbiomes of the two corals (Supplementary Table 1).

**Coral microbiomes differ in their diversity and evenness.** Bacterial diversity of *A. hemprichii* and *P. verrucosa* showed different patterns in response to anthropogenic impacts. While bacterial community richness and diversity of *A. hemprichii* responded to the degree of observed anthropogenic impacts, bacterial community richness and diversity in *P. verrucosa* was stable (Fig. 5). *A. hemprichii* had a 30% lower OTU richness (Chao estimate) at local wastewater and nutrients sites than at other sites (Fig. 5a; Kruskal–Wallis, *H* = 12.51, *p* < 0.005; pairwise tests *N* > 7.7, *p* < 0.01). Moreover, the bacterial communities at unimpacted sites (A-B) were more even than at impacted sites (Fig. 5b; Kruskal–Wallis, *H* = 15.17, *p* < 0.001), with significantly lower evenness at the local wastewater and nutrients sites and the

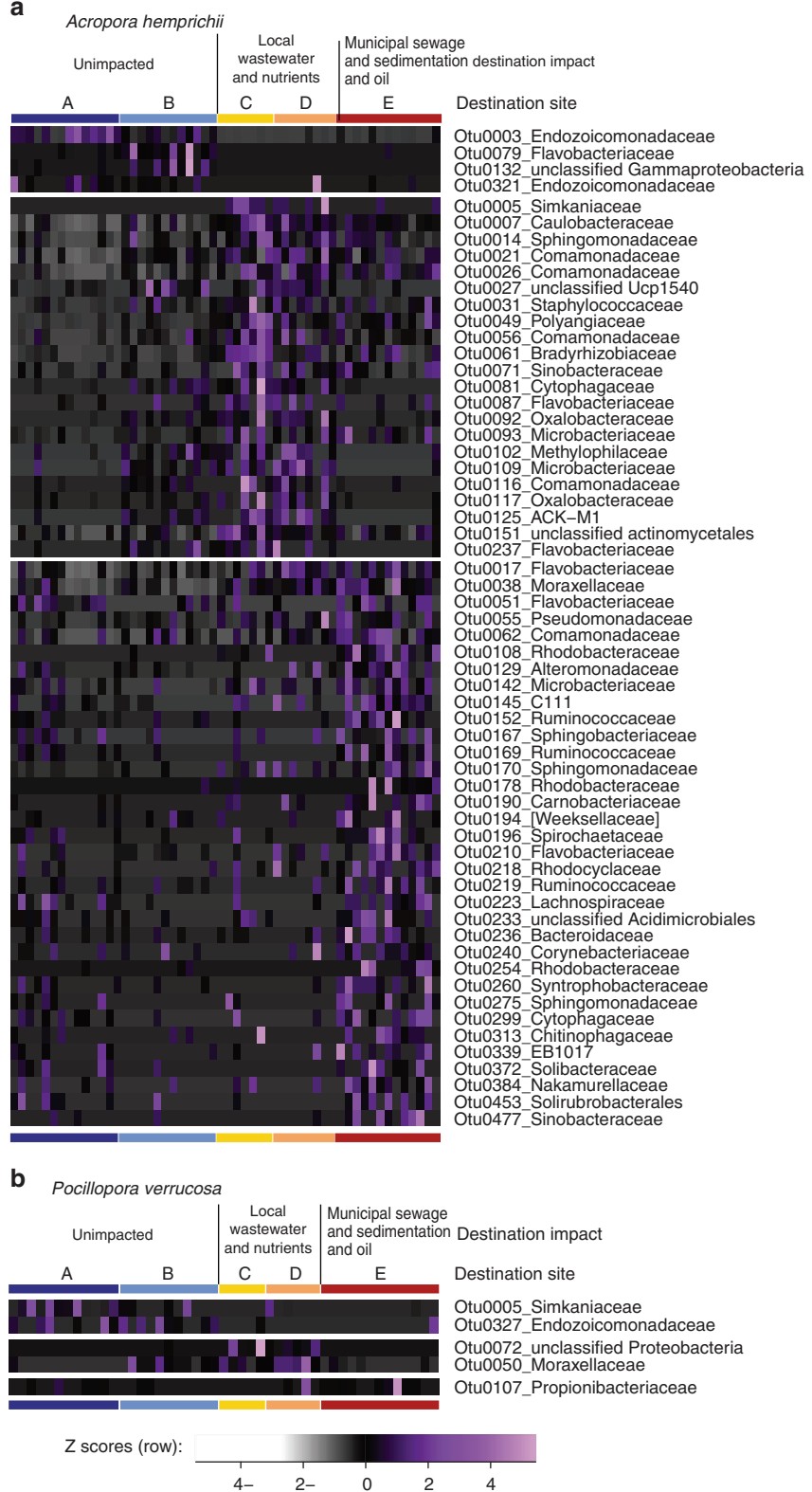

**Fig. 3** Differentially abundant bacterial taxa (OTUs) over impacts. Bacterial taxa were associated with the corals **a** *A. hemprichii* and **b** *P. verrucosa* across sites of differing anthropogenic impact near Jeddah, Saudi Arabia. The first cluster in each panel is significantly more abundant at unimpacted sites, the second cluster at sites exposed to local wasetwater and nutrients, and the third cluster at sites with municipal sewage and sedimentation and oil (LEfSe method, all LDA >2.0, $p < 0.05$). Sites and impacts marked with horizontal bars above the heatmaps. Source data are provided as a Source Data file

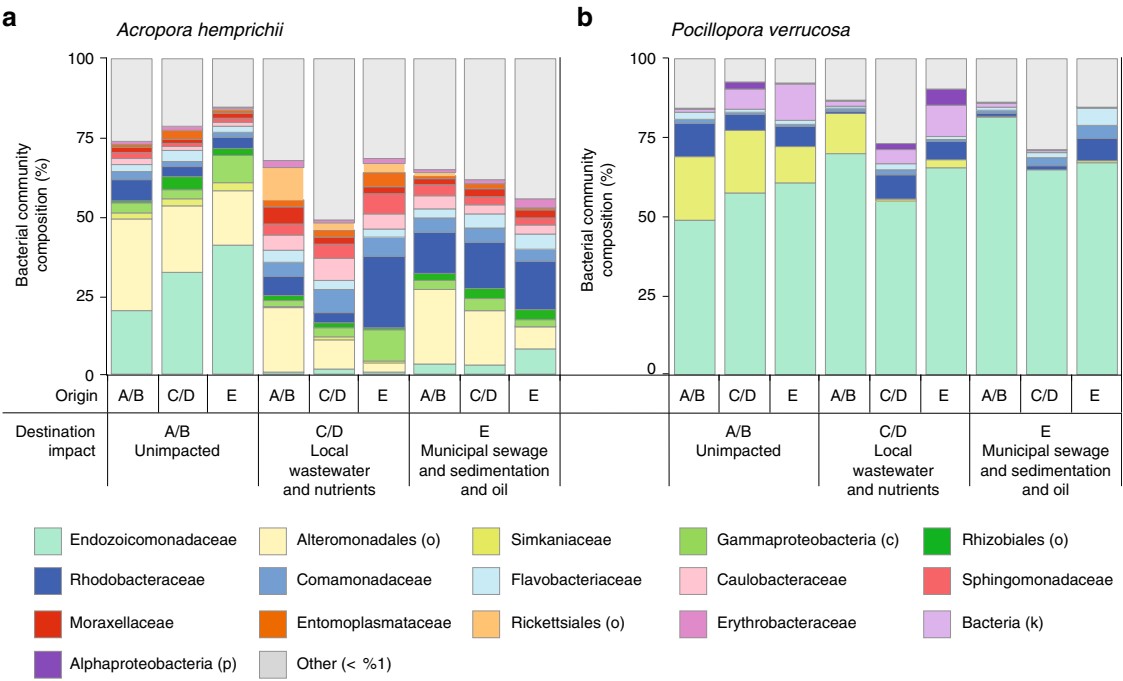

**Fig. 4** Bacterial community composition of the corals *A. hemprichii* and *P. verrucosa*. Bacterial communities shown per coral species **a** *A. hemprichii* and **b** *P. verrucosa* across sites of differing anthropogenic impact near Jeddah, Saudi Arabia. Stacked column plots display the most abundant bacterial families (>1%, determined separately for each coral species). Samples were collected after 21 months of reciprocal transplantation. A/B unimpacted sites; C/D local wastewater and nutrients sites; E municipal sewage and sedimentation and oil site. Source data are provided as a Source Data file

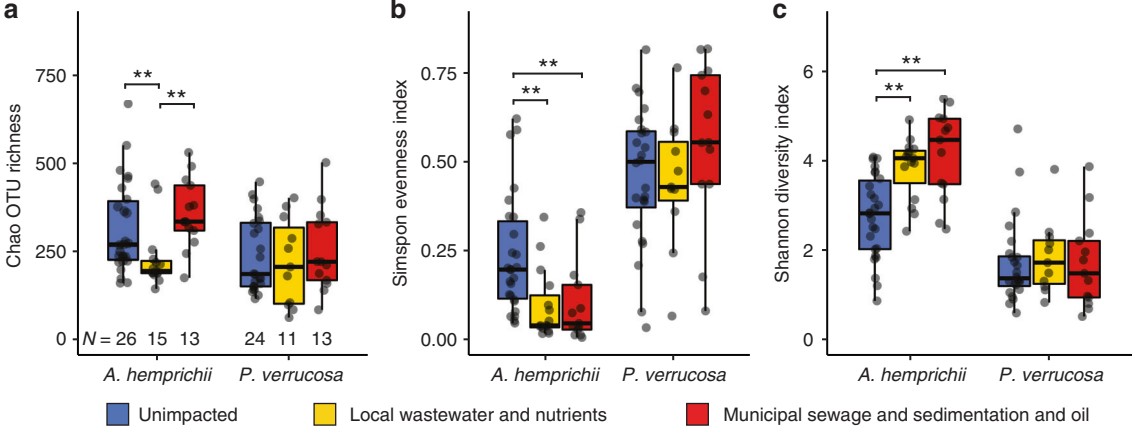

**Fig. 5** Bacterial diversity of the corals *A. hemprichii* and *P. verrucosa* across impacts. Samples were collected after 21 months of reciprocal transplantation across different impacts ("Unimpacted", "Local wastewater and nutrients", "Municipal sewage and sedimentation and oil") near Jeddah, Red Sea and are displayed by destination impact. **a** Chao Index of bacterial richness, **b** Simpson Index of bacterial community evenness, **c** Shannon Index of bacterial diversity. Boxes of box plots represent the 25th to 75th percentile, lines show medians, error bars represent smallest/largest values to a maximum of 1.5 * IQR. N of each group indicated in **a**, Kruskal–Wallis tests significance level: **$p < 0.01$. Source data are provided as a Source Data file

municipal sewage and sedimentation and oil site, compared to the unimpacted sites (pairwise tests $N = 11.63$, $p < 0.001$ and $H = 8.7$, $p < 0.005$, respectively). Consequently, overall OTU diversity (Shannon diversity) in *A. hemprichii* was significantly different between impact levels (Fig. 5c; Kruskal–Wallis, $H = 18.31$, $p < 0.001$) with significantly lower diversity at unimpacted sites compared to local wastewater and nutrients sites (pairwise test $N = 12.57$, $p < 0.001$), and the municipal sewage and sedimentation and oil site (pairwise test $N = 11.33$, $p < 0.001$). In contrast, differences in bacterial diversity between impacts were minor in *P. verrucosa* (Fig. 5). Overall, OTU richness in *P. verrucosa* was similar between environmental impacts (Fig. 5a; Kruskal–Wallis,

$H = 0.58$, $p > 0.05$). Similarly, community evenness in *P. verrucosa* was similar between impacts (Fig. 5b; Kruskal–Wallis, $H = 3.48$, $p > 0.05$). Further, community evenness was much higher than for *A. hemprichii* (Fig. 5b), resulting in overall low bacterial diversity estimates (Fig. 5c; Kruskal–Wallis, $H = 0.57$, $p > 0.05$). Notably, re-analyses of the dataset at a 99% OTU similarity cutoff recaptured the observed patterns (Supplementary Fig. 9, Source Data). However, it should be noted that because the Endozoicomonadaceae are a dominant feature in the *P. verrucosa* microbiome, excluding bacteria of this family changed the evenness and diversity of these microbial communities (Supplementary Fig. 9; Source Data).

## Discussion

Changes in microbial community structure represent a potentially fast and flexible mechanism that may facilitate coral holobiont adaptation and broaden plasticity[1,31,32]. In this study we tested two coral species for their capacity to undergo microbiome restructuring in response to changing environmental conditions during a long-term reciprocal transplantation experiment. We found that the *A. hemprichii* microbiome is highly flexible and more variable, whereas the *P. verrucosa* microbiome is fairly stable and overall less variable in response to changing environmental conditions. These findings suggest that coral species exhibit different degrees of flexibility in holobiont structure and composition.

*A. hemprichii* underwent strong microbiome restructuring when transplanted from unimpacted sites to impacted ones. This was characterized by increased bacterial diversity and decreased evenness (i.e., the loss of dominant taxa from the unimpacted site). An increase in bacterial diversity in coral microbiomes often accompanies the holobiont stress response as a result of emerging opportunistic taxa that are otherwise absent or suppressed[33,34]. Also, increased bacterial diversity has been repeatedly observed in diseased coral microbiomes[35,36]. Notably, the changes in bacterial communities of *A. hemprichii* at impacted sites were characterized by decreased relative abundances and potential loss of Endozoicomonadaceae, the bacterial family containing the enigmatic coral symbiont genus *Endozoicomonas*[37,38]. Moreover, bacterial communities of *A. hemprichii* fragments transplanted to impacted sites had higher abundances of bacterial families that have previously been characterized as opportunists and that have been associated with coral disease. For instance, taxa within the Rhodobacteraceae have been found on corals with white plague disease[35] and the family Flavobacteraceae also contains potentially pathogenic taxa[36]. Nonetheless, other bacterial families that were more abundant in *A. hemprichii* at the impacted sites (Erythrobacteraceae, Comamonadaceae, Oxalobacteraceae, Moraxellaceae) have also been isolated from healthy corals, which illustrates that shifts in microbial abundances do not exclusively or necessarily reflect a pathobiome, but rather an environmentally selected, putatively more beneficial microbiome[39–41]. Indeed, the notion of an environmentally explicit set of bacterial taxa that fill specific functional niches through changes in the microbiome of *A. hemprichii*[17] is supported by the high number of specific taxa found under different impacts in our LEfSe and indicspecies analyses. In contrast, the number of such environmentally explicit taxa in *P. verrucosa* was an order of magnitude lower (5 and 7 taxa in *P. verrucosa* vs. 60 and 62 taxa in *A. hemprichii* for the indicspecies and LEfSe analyses, respectively). This is also in line with the current notion that holobiont composition (or metaorganism structure for that matter) is not static, but rather dependent on age, development, sex, and environment, among other factors[42]. As such, consistent or close association of bacteria with their animal hosts is not a *sensu stricto* criterion for functional relevance[42]. Furthermore, the microbiome of *P. verrucosa* was remarkably consistent between sites, with only small changes between impact levels in bacterial diversity and evenness. The most abundant Endozoicomonas OTU in *P. verrucosa* dominated most samples and, in contrast to the loss of Endozoicomonadaceae in *A. hemprichii* at impacted sites, was consistent. Generally, there were no major abundance changes in bacterial taxa between impact levels in *P. verrucosa*, with the exception of the Simkaniaceae family, whose role we can only speculate on. Additional analyses excluding the Endozoicomonadaceae family further support the notion of a more stable and less variable microbiome of *P. verrucosa* compared to *A. hemprichii*. Whether dominance of a particular bacterial lineage promotes a generally more stable microbiome (inducing stability

of less abundant members), or whether it is simply the indication of an inflexible host-microbial association, remains to be determined.

The differences we observe in bacterial assemblages between *A. hemprichii* and *P. verrucosa* across different sites argue for different degrees of microbiome flexibility. Thus, these potentially represent differences in the underlying strategy employed by the two species to cope with environmental stress. Importantly, it has to be considered that corals are in general long-lived, sessile animals that are unable to escape changes in their environment. And because of their long generation times, evolutionary change is supposedly slow[43]. Strategies to cope with and survive rapid environmental change are therefore critical. One mechanism by which corals may adjust more rapidly to change may be through their association with different bacterial taxa, whereby selection occurs for the most advantageous and beneficial microbiome in a particular environment[1,10]. It is therefore striking to find in *A. hemprichii* a readily "responding" microbiome, in contrast to which the bacterial community of *P. verrucosa* seems rather "inert". However, this finding is not entirely unexpected. Previous studies have shown highly stable Symbiodiniaceae[26,44] and bacterial[14,24,28] communities both in *P. verrucosa* and also its close relative *Pocillopora acuta*[45], even under conditions of mortality. In comparison the microbial communities of *Acropora* are flexible and seem to align with environmental patterns[13,29,46]. In line with these studies, we found the bacterial microbiome of *A. hemprichii* to be highly variable between sites and impacts and also flexible upon reciprocal transplantation. In contrast, *P. verrucosa* harbored a far less variable bacterial microbiome that showed less flexibility upon transplantation.

Consequent to these findings, we argue that coral species differ in their ability to associate flexibly with different bacterial assemblages. In analogy to strategies for coping with osmotic stress[47], *Acropora* might be referred to as a "microbiome conformer" (showing microbial adaptation to the surrounding environment), whereas *Pocillopora* might be referred to as a "microbiome regulator" (showing microbial regulation that maintains a constant microbiome). Hence we propose the term "microbiome flexibility" to describe a coral species' potential for dynamic microbiome restructuring in the face of environmental change. It remains to be investigated how pervasive these patterns of microbiome flexibility are across and within coral taxa, as illustrated by instances of reversed patterns of relatively high microbiome flexibility in *Pocillopora*[41] and relative low microbiome flexibility in *Acropora*[27] (although each of the studies tested only one species separately, rendering cross-species comparisons difficult). It further remains to be determined whether differences in microbiome flexibility represent distinct holobiont adaptation mechanisms to environmental change. High microbiome flexibility presumably supports holobiont adaptation to environmental change and follows a generalist strategy albeit with the risk of losing important associates/functions or acquiring pathogens. Conversely, low microbiome flexibility helps to maintain stable and robust relationships with conserved microbial functions reflecting a more specialized strategy at the expense of a putatively low capacity for microbiome adaptation and potential susceptibility to rapid environmental change.

The degree of microbiome flexibility may be linked to life history strategy of the host[48]. *A. hemprichii* and *P. verrucosa* both have wide distribution ranges[49,50] and inhabit the shallow to mid reef slope[50]. However, the microbiome conformer *A. hemprichii* has a relatively slow growth rate[51] and, in contrast to Pocilloporids, Acroporids generally have a competitive strategy, with longer generation times[52]. Indeed, the microbiome regulator *P. verrucosa* shows a rather limited physiological plasticity and ecological niche space, being adapted to high-light wave-exposed

environments near the reef crest, in particular in the Red Sea[44]. Pocilloporids generally favor an opportunistic colonization strategy that is characterized by fast growth, high reproduction rates, and relatively rapid generation times[53]. Given that a limited diversity in the diet of deep-sea corals has been linked to reduced diversity within the microbiome[48], the mainly autotrophic lifestyle and low heterotrophic capacity of *P. verrucosa* may also partially explain its low microbiome flexibility[48]. However, it remains to be determined whether low microbiome flexibility is a cause or consequence of this strategy.

The presence of varying degrees of microbiome flexibility among different coral taxa may also help to resolve the discrepancy regarding the absence of a consistent coral core microbiome. While the concept of a core microbiome may be debatable, especially in the light of the coral probiotic hypothesis[1,10], the expectation is that at least some bacterial associates are intimately and tightly associated with a coral and not expendable[14,17,54,55]. A recent study[54], after examining two coral species across a wide range of habitats, proposed seven distinct bacterial phylotypes as universal coral core microbiome members. Surprisingly, these core OTUs were nevertheless not ubiquitous, the threshold used to qualify as a core microbiome member being comparably low at ≥30% of samples. Our data suggest that attempts to identify a universal coral core microbiome might result in only a small number of bacterial taxa, if different coral species are compared that display such high or low microbiome flexibility. Potentially relevant here is that coral phylogeny is characterized by a deep phylogenetic split (between "complex" and "robust" clades) that dates back >245 mya[56,57]. This led to substantial genomic divergence[58] and consequences for host-microbe pairings that may be shaped by phylosymbiosis[59]. Such a separation might arguably underlie not only differences in microbial association, but also differences in microbiome flexibility, such as observed here.

A final point arising from results of cross-transplantation and back-transplantation is that the bacterial communities of both coral species were similar to their local back-transplanted conspecifics after cross-transplantation to unimpacted sites. This finding may hold the promise of microbiome "recovery". In other words, coral fragments transplanted from impacted to control sites did not, after 21 months, continue to share similarities with fragments of the same colonies that remained at the impacted sites. These findings suggest that stress-induced microbiome alterations may be reverted upon removal of chronic and long-term stressors, similar to the recovery observed after coral bleaching[29] or disease[30]. Following the notion that coral microbiomes contribute to coral health, our results indicate that reducing and removing sources of pollution and sedimentation may result in the reversal of microbiomes. Hence, anthropogenic pollution may not irreversibly disrupt microbiomes in supporting coral health[1,60,61]. Our results are in line with recent studies, which report that increases in coral disease caused by experimental nutrient enrichment were reversed 6–10 months after termination of the experimental treatment[62]. Taken together, our data create an additional incentive to reduce sources of anthropogenic pollution and sedimentation close to coral reefs, even if the corals on the target reefs already appear stressed and in poor condition.

Changes in the microbiome of plant and animal hosts are increasingly being associated with the potential for acclimatization and adaptation in multicellular organisms. In particular, stony corals seem to be strongly reliant on their microbial associates, as highlighted by their obligate endosymbiosis with algal dinoflagellates. However, the potential for differences in microbiome flexibility across coral species had not until now been systematically investigated. Our study supports the notion that distinct degrees of microbiome flexibility exist, potentially reflecting different holobiont adaptation mechanisms to environmental change. Thus, bacterial community structures may respond in a host-specific manner (not "one-size-fits-all"), a situation which would hamper elucidation of a universal core microbiome. Importantly, altered microbial community structures of corals from impacted sites recovered, being indistinguishable from local conspecifics when transplanted back into an unimpacted control environment. This finding holds the promise of microbiome recovery, encouraging the reduction of anthropogenic pollution, even in reef areas where coral assemblages are already degraded.

## Methods

**Experimental design and sample collection**. Six study sites (A–F) were selected as described in the previous work of Ziegler et al.[28] (Fig. 1). The experimental setup at site F was destroyed during the experiment and hence results from five sites are reported hereafter (A–E). These sites represent a range of anthropogenic impacts with differences in abiotic factors, benthic community composition, seawater microbial communities, and coral-associated bacterial community compositions.

Sites A and B were relatively unimpacted and represent almost pristine control conditions. Both locations were characterized by comparatively low sedimentation loads (Supplementary Fig. 10, Source Data), low inorganic nitrate concentrations (Supplementary Fig. 11, Source Data), and low levels of total hydrocarbons (THC), measured against a standard of Light Arabian Crude Oil (Supplementary Fig. 12, Source Data; see Supplementary Methods for details on measurements). During previous surveys a high stony coral cover and diversity as well as low abundances of soft corals was recorded at these sites[28]. Sites C and D were located along the fringing reef of the heavily developed Jeddah Corniche and represent an intermediate impact level. The area is exposed to chronic turbidity and intermediate sedimentation loads from infilling (Supplementary Fig. 10, Source Data) paired with unauthorized local wastewater outfalls that are estimated to release 99,000 $m^3\,d^{-1}$ of untreated wastewater into the nearshore area along 30 km of coastline and lead to elevated nitrate levels (Supplementary Fig. 11, Source Data)[63,64], while levels of THC are comparable to the unimpacted sites (Supplementary Fig. 12, Source Data). Both sites were characterized by relatively high cover of *Xenia* spp., known to opportunistically invade degraded reefs[65]. Site E represents the most severe impact level being located within Jeddah Bay, in proximity to the industrial port, which generates intermediate levels of oil pollution (Supplementary Fig. 12, Source Data). In addition, site E was less than five km from the three main discharge points of Jeddah's sewage treatment facilities, which regularly discharge extensive amounts (35,000, 68,000, and 300,000 $m^3\,d^{-1}$, respectively) of untreated or only partially treated sewage that lead to intermittent increases in nitrate levels (Supplementary Fig. 11, Source Data)[66,67]. Hence, this site is subjected to elevated turbidity and highest sedimentation loads (Supplementary Fig. 10, Source Data). Hard coral cover at site E is similar to sites C and D with soft coral cover being intermediate, with lower counts of *Xenia* spp.[28].

In total, 36 coral colonies (18 per species) were each fragmented into 6 pieces resulting in 216 coral fragments that were reciprocally transplanted. Specifically, at each of the six initial sites (A–F), three visually healthy colonies of each of *A. hemprichii* and *P. verrucosa* were collected at 5–10 m depth, fragmented into six or more pieces (about 10 cm max length) each, and suitable pieces then transported to the King Abdulaziz University marine laboratory at Sharm Obhur for measurement and weighing. The fragments were allowed to recover for 2–3 weeks at a depth of 5 m deep on a shallow reef adjacent to the laboratory, and then cross- and back-transplanted across all sites during July/August 2013 (reciprocal transplantation design, Fig. 1). This design allowed fragments from each colony to be assessed at each location, including one fragment per colony that was back-transplanted to its respective site of origin. Of note, the back-transplanted fragments (to their site of origin) act as experimental and handling "controls" in that any microbiome changes would arguably be a result of the transplantation procedure. These fragments also aid in correcting for possible changes of the microbiome that may be related to time or age of the fragments, in that fragments from the same colonies (with the same age, sharing the same life history) were investigated at all sites[68]. Coral fragments were attached by cable-ties to a frame suspended in mid-water at 5–10 m depth to prevent sediment cover and predation by benthic predators. Experiment sites were then visited approximately every three months to monitor growth and check the integrity of the structures. Coral fragments were retrieved after a further 21 months (April/May 2015), when they were rinsed with filtered seawater, placed individually in sterile polyethylene bags, and transported back to the Sharm Obhur laboratory on ice, before being frozen for temporary storage at −80 °C. Using cubitainers, four water samples (1 L) were also collected from the same depth as the coral samples at each site, and likewise transported in dark portable coolers back to Sharm Obhur, where they were filtered using 0.22 μm Durapore PVDF filters (Millipore, Billerica, MA, USA). Coral samples and water filters were transported frozen to King Abdullah University of Science and Technology (KAUST) and kept at −80 °C until further analysis.

**DNA extraction and 16S rRNA gene amplicon sequencing.** Between 2 and 4 ml PBS were applied to each frozen coral fragment before tissue was sprayed off into sterile zip lock bags using sterile filter pipette tips (1000 μl barrier tips, Neptune, USA) connected via a rubber hose to a bench top air pressure valve. 0.5 M EDTA was added to the resulting coral tissue slurry to a final concentration of 1.25 mM before storage at −80 °C. DNA was extracted with the DNeasy Plant Mini Kit (Qiagen, Hilden, Germany) following the manufacturer's protocol and using a volume of 100 μl coral tissue slurry suspended in 300 μl AP1 lysis buffer. For DNA extraction from water samples, each filter was cut into three pieces using sterile surgical blades. One piece was sliced into 2–4 mm wide strips, suspended in 400 μl AP1 buffer in a 1.5 ml centrifuge tube, and incubated for 30 min on a rotating wheel at a 45° angle. The DNeasy Plant Mini Kit protocol was then used to extract DNA. DNA concentrations of all samples were measured using a NanoDrop 2000C spectrophotometer (Thermo Fisher Scientific, Waltham, MA, USA).

For PCR reactions between 5 and 50 ng template DNA was used for coral samples and between 4 and 10 ng template DNA for water samples. PCRs were performed in triplicate 10 μl reactions using the QIAGEN Multiplex PCR kit and final primer concentrations of 0.5 μM. Variable regions 5 and 6 of the 16S rRNA gene were amplified using primers 784F [5′TCGTCGGCAGCGTCAGATGTGTA TAAGAGACAGAGGATTAGATACCCTGGTA′3] and 1061 R [5′GTCTCGTGG GCTCGGAGATGTGTATAAGAGACAGCRRCACGAGCTGACGAC′3][69] with Illumina adapter overhangs (underlined above). The PCR conditions consisted of initial denaturing at 95 °C for 15 min, followed by 27 cycles of 95 °C for 30 s, 55 °C for 90 s, and 72 °C for 30 s, and a final extension step at 72 °C for 10 min. PCR amplification was visually confirmed using 1% agarose gels with 5 μl PCR product. The triplicate PCR amplicons were pooled, 5 μl of each pooled sample was aliquoted, and samples were then cleaned using illustra ExoProStar 1-step (GE Healthcare Life Sciences, USA). Indexing was performed using Nextera XT indexing adapters (Illumina, USA). Samples were then cleaned and normalized (to 25 ng DNA per sample) using the Invitrogen SequalPrep Normalization Plate Kit (Thermo Fisher Scientific, USA). Normalized samples were pooled and sequenced at the KAUST Bioscience Core Lab (BCL) on the Illumina MiSeq platform using 2 × 300 bp paired-end v3 chemistry at 8 pM with 10% PhiX.

**Sequence data processing and analysis.** Sequence data processing and analysis were done using mothur v. 1.41.1[70]. Specifically, after forward and reverse reads were assembled into contigs, ambiguous reads were removed, and sequences were quality trimmed and pre-clustered[71]. Singletons were discarded and the remaining sequences were aligned against the SILVA database[72]. Chimeric sequences were excluded using VSEARCH[73] and chloroplast, mitochondrial, archaeal, and eukaryotic sequences were removed. The remaining sequences were then classified against the Greengenes database[74]. Three bacterial taxa that comprised >95% of sequences in the DNA extraction negative control and the PCR negative control were removed from all samples for subsequent analyses using the remove.lineages() command in mothur. These taxa were an unclassified *Brachybacterium* in the family Dermabacteraceae, an unclassified *Dietzia* in the family Dietziaceae, and the bacterium *Brevibacterium casei*. The latter two have been reported previously as common laboratory contaminants[75]. After removal of putative contaminants, samples were subsampled to 3,101 sequences and again classified against the Greengenes database. Stacked column plots representing bacterial community compositions at the taxonomic family level were constructed based on these phylogenetically classified sequences. For all subsequent analyses, Operational Taxonomic Units (OTUs) were built based on sequence clustering at a 97% similarity cutoff. Because this cutoff can be considered rather conservative, a comparative analysis of a second dataset using a 99% similarity cutoff for OTU clustering was also conducted. This analysis yielded similar sample groupings and results (Supplementary Data 3–4). Given that the microbiome of *P. verrucosa* was dominated by a single bacterial taxon (OTU0001, Endozoicomonadaceae), a third dataset was created that excluded the Endozoicomonadaceae family with the command remove.lineages() in mothur as detailed above. This dataset was then analyzed using the same procedures and confirmed the observed differences in microbiome flexibility between the two species (Source Data). For statistical analyses, samples were grouped by sample type and coral host species (i.e., seawater, *A. hemprichii*, *P. verrucosa*) as well as origin and/or destination sites (A, B, C, D, E) or impact levels (unimpacted—sites A, B; local wastewater & nutrients—sites C, D; municipal sewage and sedimentation and oil—site E).

All data analyses were performed in R version 3.5.0[76]; for multivariate statistics the package vegan was used[77] and for illustrations the package ggplot[78]. For multivariate analyses, OTU count data was square-root transformed. To test microbiome flexibility, the "betadisper" function was used to calculate multivariate dispersion of samples (Bray-Curtis distances) between coral species and between impacts within species. Homogeneity of multivariate dispersions were tested with ANOVAs, followed where applicable by Tukey's Honest Significant Differences post-hoc tests and visualized with boxplots.

Differences between groups were visualized using non-metric MultiDimensional Scaling (nMDS). Because dispersion of samples was significantly different between groups (function "betadisper", see above), multivariate statistical analyses were conducted per sample type (i.e., separately for seawater, *P. verrucosa*, and *A. hemprichii* samples). One-factorial PERmutational MANOVA (PERMANOVA) was run with 9,999 permutations to test for differences between

sites for seawater samples with the "adonis" function. Two-factorial PERMANOVAs were run with 9,999 permutations to test for effects of origin and destination sites for each coral species separately in a fully crossed design. This analysis was repeated with pooled sites per impact. The Linear discriminant analysis Effect Size (LEfSe) method[79] was used with Kruskal–Wallis and Wilcoxon tests to identify shifts in the abundance of OTUs between impacts ($p = 0.05$; logarithmic LDA threshold = 2) as implemented in mothur.

Alpha diversity indices (Chao1, Simpson Evenness, Shannon Diversity) were calculated for the bacterial community of each sample in mothur and analyzed for differences between destination impact levels using Kruskal–Wallis tests in R. Bacterial indicator taxa analyses were conducted to identify bacteria that associate with corals at specific impact destinations and to investigate whether corals originating from the same impact continue harboring specific taxa even after transplantation (*p*-value threshold = 0.01). The analysis was conducted in R using the indicspecies package with the command "multipatt"[80]. We also investigated the members of the microbiome that were broadly present and presumably stably associated with *A. hemprichii* and *P. verrucosa*. Such putative core microbiome members were determined for each species separately by querying all bacterial OTUs that were present in ≥75% of samples at each site, ensuring even distribution of these core taxa across the sample set[55].

**Reporting summary.** Further information on research design is available in the Nature Research Reporting Summary linked to this article.

## Data availability

Sequence data determined in this study are available under NCBI BioProject ID PRJNA491299. Abundant coral bacterial microbiome OTU reference sequences are available under GenBank accession numbers MK736129 to MK736265 for *Acropora hemprichii* and under GenBank accession numbers MK736048 to MK736100 for *Pocillopora verrucosa*. Source data underlying figures and statistical analyses are provided as a Source Data File.

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

## Acknowledgements

We would like to thank Adam Porter (University of Exeter, UK) for assistance with the fieldwork and Craig Michell (King Abdullah University of Science and Technology, KAUST) for help with sequencing library preparation. We acknowledge the KAUST Bioscience Core Lab for assistance with MiSeq sequencing. Research reported in this publication was supported by baseline funds to CRV from KAUST and undertaken as part of the Chair's Program in Coastal Marine Conservation at King Abdulaziz University, Jeddah, funded by HRH Prince Khaled bin Sultan. The research was further supported by scholarships under the KAUST visiting student research program (VSRP) to C.G.B.G. and M.M.B.

## Author contributions

M.Z., R.O., and C.R.V. designed and conceived the experiment; M.E., J.B.O., A.A.S., and R.O. collected and transplanted the samples; A.J.T. collected environmental data; K.Z., A.A.S., and C.R.V. provided reagents/tools; C.G.B.G., M.Z., and M.M.B. generated the molecular data; M.Z., C.G.B.G., M.M.B., and C.R.V. analyzed the data; M.Z., C.R.V., and C.G.B.G. wrote the manuscript which R.O. edited; M.Z., C.R.V. revised the manuscript with input from R.O., C.G.B.G.; all authors read and approved the final manuscript.

## Additional information

**Competing interests:** The authors declare no competing interests.

