## [Peer Review File · Nature Communications]

Reviewers' comments:

Reviewer #1 (Remarks to the Author):

In their manuscript entitled "Differential bacterial restructuring in coral species upon transplantation to adverse environmental conditions provides insight into microbiome flexibility", Grupstra and colleagues describe a study investigating the potential for two coral species, *Acropora hemprichii* and *Pocillopora verrucosa*, to restructure their associated bacterial communities when transplanted to different localized level of anthropogenic impact. The authors have employed this existing gradient in environmental impact to good affect in previous work, including studies with both *A. hemprichii* and *P. verrucosa*. While coral microbiome studies are frequently plagued by relying on more promise than actual, meaningful insight into coral holobiont biology or ecology, this study avoids that failing. Overall, this is a very nicely designed study that has also resulted in a well-written manuscript.

The microbiome flexibility idea is intriguing, and supported by other work from Voolstra and colleagues as well as others. However, I have some concerns that confounding factors exist that are not addressed when trying to interpret the difference in how the microbiomes of *A. hemprichii* and *P. verrucosa* respond to transplantation in the manner done in this manuscript. This gets to the core of the "differential bacterial restructuring" highlighted by the manuscript title. Because *P. verrucosa* has a microbiome dominated by a single *Endozoicomonas* lineage, this necessarily forces two things: first, it highly skews the structure of the *P. verrucosa* microbiome, greatly impacting both alpha and beta diversity and the statistical analyses applied to each. Second, it is very likely the case that comparing the microbiomes of these two corals is essentially akin to comparing apples and oranges. It is known that *Endozoicomonas* are generally found inside the tissues of the coral animal, and has such should not be expected to respond to environmental stimuli in the same manner as microorganisms residing exterior to the coral animal tissue, or respond on the same temporal scale. Is it particularly meaningful to compare a microbiome known to be dominated by an intracellular monoculture to one where the location of microbial cells is not known, but is probably dominated by cells that reside on the exterior of the coral tissue?

Specific points:

Lines 52-54: no experiments were performed to assess whether a change in microbiome is a strategy for coping with environmental change. This is purely conjecture.

Lines 218-219: 97% cutoff for OTUs is now considered quite conservative, and possibly masks some interesting dynamics.

Lines 274-285: It is interesting that no abundant cyanobacteria were recovered from the seawater samples. Why is that?

Line 337: Replace "3C" with "3B".

Reviewer #2 (Remarks to the Author):

The study from Grupsta et al. is investigating microbiome changes in response to anthropogenic impacts in two different coral species. The author cross-transplanted coral fragments from 5 sides for the duration of 21 month. At the end of 21 months microbiomes were analyzed based on 16S rRNA gene analysis. The authors claim that the two coral species show different "microbiome flexibility". While one coral responded with large changes in the microbiome, the other species was more stable in its microbiome.

I have two major points that should be considered:

1) The authors argue, that the coral P.v. is less flexible in its microbiome compared to the coral A.h. In Figure 2A all analyzed samples from both corals are depicted. If the interpretation is right, that A.h. is more flexible compare to P.v., then the within treatment distances calculated based on the distance matrix should be significantly smaller in A.h. compared to P.v. Just based on Figure 2A, I would not expect that these values are different.

In addition, I would expect that in the less flexible coral P.v. "origin site" should explain some of the observed microbiome differences. I agree that the changes in A.h. can be better explained by the destination site, as in P.v. But the general microbiome flexibility seem to be as high in P.v. as in A.h.. I encourage the authors to address this difference, by differentiating between general microbiome flexibility, which seem to be similar between both species (based on Figure 2a), and differences, which are explained by destination site.

One could even argue other way around. The microbiome of A.h. responds to destination side more robust compared to P.v.. In P.v. the microbiome accumulates more stochastic differences over time, that are not explained by destination side, and therefore is more flexible.

2) The authors state in the abstract (line 49) and in the discussion (line 423) that back-transplants were performed to measure the potential of the microbiome to recover. But in the M& M section it is written that only one cross-transplantation experiment was performed. For a back-transplantation experiment, I would expect a second transplantation after the first 21 month. In addition, I miss a corresponding data analysis in the results section.

Specific points:

Figure 2A: Are water samples taken from all sides? I would expect, that anthropogenic effects should be reflected also in the water microbiome, as also stated by the authors in line 119. But in Figure 2A all samples are highly similar. Which biotic and abiotic factors are known to be different at the different sides? Temperature, salinity, DOC, etc? The contribution of measured environmental data should be added to the analysis in addition to destination side, which is not well-defined.

Figure 2B,C: In Table 2 it is shown that also the comparison of destination sides A vs.B are significantly affecting microbiome structure. Therefore, I would suggest coloring all destination sides with unique colors throughout the manuscript.

Line 354: Based on Figure 4 the authors argue that there is a qualitative difference in the abundance changes between A.h. and P.v.. In my opinion, bar charts are not the adequate method to illustrate that. I encourage the authors to show the LEfSe analysis e.g. by using LefSe cladograms. The qualitative differences in the responsiveness between A.h. and P.v. do not get evident in the bar charts.

Reviewer #3 (Remarks to the Author):

The paper by Grupstra et al aiming to explore microbiome flexibility in corals is certainly of interest and an exciting experiment with lots of promise. However, as its currently written I do not see it being published in this journal. That said I would be more than happy to see a revised version answering or exploring some of the issues I identify below.

Firstly, I think the paper is possibly over stating its novelty. Two similar (but of course less detailed) studies were published a while back by Garren et al. in 2008 and 2009

Garren, M., Raymundo, L., Guest, J., Harvell, C. D., & Azam, F. (2009). Resilience of coral-associated bacterial communities exposed to fish farm effluent. PLoS One, 4(10), e7319.

Garren, M., Smriga, S., & Azam, F. (2008). Gradients of coastal fish farm effluents and their effect on coral reef microbes. *Environmental microbiology*, 10(9), 2299-2312.

Which may be worth exploring. I should note at this point just cause I highlight a paper does not mean you have to cite it, just please read it and see if it fits with what you are saying.

Furthermore, a couple of studies by Sweet et al. show both short term mechanical disruption and recovery of the microbiome and more natural variation over tidal stress – one which you rightly cite but could explore in more detail maybe. Also from the same group they highlight how the microbiome varies with age, this might be interesting to at least note and discuss later and its implications for your study and the age of your corals sampled.

Sweet, M. J., Croquer, A., & Bythell, J. C. (2011). Dynamics of bacterial community development in the reef coral *Acropora muricata* following experimental antibiotic treatment. *Coral Reefs*, 30(4), 1121.

Sweet et al (2013). Changes in microbial diversity associated with two coral species recovering from a stressed state in a public aquarium system. *Journal of Zoo and Aquarium Research*, 1(2), 52.

Williams, A. D., Brown, B. E., Putschim, L., & Sweet, M. J. (2015). Age-related shifts in bacterial diversity in a reef coral. *PLoS One*, 10(12), e0144902.

Introduction

I think the intro could be a little more topical and inclusive of the current literature, some highlighted above. Furthermore you should rename Symbiodinium in accordance with the new classification which at least one of the authors of this publication was involved in ;)

Referencing throughout also needs to be addressed for example L73 but quite a few other examples and some mistakes in the ref list.

L85 enjoy is a little to human feelingy – maybe exhibit or show

Finally a third of your intro is put over to what you did – usually this is only a small part highlighting your main aim and I feel you make more use of the literature to tell a stronger and more compelling story about why you need to do what you have done

For example you explore what we know about the corals structuring of the microbiome, see Sweet et al 2011 and a more recent paper by Pollock which I believe just came out

<https://www.nature.com/articles/s41467-018-07275-x>.

Sweet, M. J., A. Croquer, and J. C. Bythell. "Bacterial assemblages differ between compartments within the coral holobiont." *Coral Reefs* 30.1 (2011): 39-52.

Materials and Methods

My first question was where is F but then you explain it got wiped out, I might suggest therefore to remove it completely as you give it no further consideration afterwards inc in figure 1

You mention differences in benthic community, seawater microbes etc, are these highlighted in Zieglers paper? If so say so or maybe make a supplementary table

L156 swab approx. and every around

L146-155 I worry about your level of replication – I started to have concerns here and as I read more and more they amplified see later comments. if I understand correctly you only have 5 frags and 5 sites. So that means one frag per site and 3 in total as you had three different colonies. My point is with the level of variation seen in the microbiome of corals in close proximity to each other, even in a healthy state this can compound your results and I think if you presented your data in a way which I will suggest later this would do exactly that.

Furthermore, did you also take a sample from the 'parental' colony before taking the colony and fragging it –you could have then compared your back transplanted sample to this to explore any observable change – which I imagine there very well may have been

L206 some worry about excluding chloroplasts – I'm in two minds but might be worth thinking about this

L2015 – I got a little confused why you started classifying with Silva then moved to greengenes?

Great that you explore core or stable microbiomes but there are plenty of issues with this and you have possibly fallen into one of the traps highlighted by Sweet and Bulling 2017

Sweet, M. J., & Bulling, M. T. (2017). On the importance of the microbiome and pathobiome in coral health and disease. *Frontiers in Marine Science*, 4, 9.

It might be worth running your analysis with 90% and 100% - in my opinion 75% is a bit of a

random number and suggests you either have a lack of confidence in your sample collection and prep or of your sequencing if you use any cut off to describe a core microbe. But this is obviously just an opinion.

Results

Table 1 was exactly why I was worried about your sample strategy and the loss of fragments – therefore I feel that your conclusions are being stretched from relatively thin data

L297 I'm not at all surprised by this as arguably a huge percentage of the corals 'microbiome' are transients – see above review and likely driven by the environment, whilst the core, possibly truly functioning microbes will not be and this is where the real interest and shifts and changes lie in my opinion

Furthermore, in Fig 2 how do we know the origin of the samples? I think it would be really useful to present these results i.e. the original coral microbiome profile, the back transplanted one, and then the change that frag goes through when transported to site B, C,D etc

Table 2, if I understand it means some of your conclusions may well be slightly off mark. Are the horizontal A,B,C etc the origin site? If so please highlight this. So A frags showed no change when transplanted to sites, B, C, D or E if I read this table this way. So in the discussion when you state 'corals microbiomes recover to their original state when local sources of pollution are removed'

L424. If true then this is what Garren found in the earlier papers so please site, but my interpretation of the table you present suggests something different. .e. if this was true I would expect corals from E transferred to A or B to show no significance in their microbiome profiles but in your table 2 they are the opposite. This got me very confused, so I stopped reading the discussion at line 428 and would recommend a revision of the work and resubmission depending on what the other reviewers and editor suggest.

That said the results from lines 321 were certainly more interesting than the previous sections (again in my opinion) – I would however err on the side of caution when identifying bacterium to species using a max of 277 bp fragments – I obvs can't check these but where there really no their blast matches close to these species?

Finally would it not be interesting in Figure 4 to keep A, B, C and D separate and this would allow within 'treatment' 'ecosystem state' visual analysis to be done by the reader and/or more reliable cross analysis with site E which you could not combine with F for obvious reasons

NCOMMS-18-29154 R1

Differential bacterial restructuring in coral species upon transplantation to adverse environmental conditions provides insight into microbiome flexibility

Dear Reviewers,

We provide detailed point-by-point responses to each of the reviewer's points in blue and highlight changes to the manuscript in green (which are also pointed out in the track-changes version of the manuscript file).

Thank you for your time.

Sincerely,
Christian

Corresponding author, on behalf of all authors

Reviewers

Reviewer #1 (Remarks to the Author):

In their manuscript entitled “Differential bacterial restructuring in coral species upon transplantation to adverse environmental conditions provides insight into microbiome flexibility”, Grupstra and colleagues describe a study investigating the potential for two coral species, *Acropora hemprichii* and *Pocillopora verrucosa*, to restructure their associated bacterial communities when transplanted to different localized level of anthropogenic impact. The authors have employed this existing gradient in environmental impact to good affect in previous work, including studies with both *A. hemprichii* and *P. verrucosa*. While coral microbiome studies are frequently plagued by relying on more promise than actual, meaningful insight into coral holobiont biology or ecology, this study avoids that failing. Overall, this is a very nicely designed study that has also resulted in a well-written manuscript. The microbiome flexibility idea is intriguing, and supported by other work from Voolstra and colleagues as well as others.

We thank the reviewer for the overall positive assessment and for their acknowledgement of the conceptual advance of our manuscript.

However, I have some concerns that confounding factors exist that are not addressed when trying to interpret the difference in how the microbiomes of *A. hemprichii* and *P. verrucosa* respond to transplantation in the manner done in this manuscript. This gets to the core of the “differential bacterial restructuring” highlighted by the manuscript title. Because *P. verrucosa* has a microbiome dominated by a single *Endozoicomonas* lineage, this necessarily forces two things: first, it highly skews the structure of the *P. verrucosa* microbiome, greatly impacting both alpha and beta diversity and the statistical analyses applied to each.

We agree with the reviewer’s concern. In a way, we have a catch-22 situation with these data, as the *Pocillopora verrucosa* microbiome may be so inflexible precisely because of the coral’s strong association with one particular bacterial lineage, compared to the more diverse and variable microbiome of *Acropora hemprichii*. It is noteworthy, however, that *Acropora hemprichii* is also strongly associated with *Endozoicomonadaceae*, but at the unimpacted sites only, indicating that the stability of the association with this bacterial lineage is host-dependent and not always an indicator for an inflexible microbiome.

In this revision, we have taken several new approaches to get to the core of this:

1. In the original manuscript and also in this revised version, the beta-diversity statistics were determined and presented on square root-transformed data. This transformation reduces the influence of more abundant members of the microbial community (such as *Endozoicomonas*) and represents a conservative approach in the data analysis. When comparing the PERMANOVA results from the untransformed with the transformed dataset, we essentially find that results stay the same, namely that the microbial community of *P. verrucosa* remained more stable across impacts than that of *A. hemprichii*, suggesting that the role of the dominant *Endozoicomonas* lineage(s) in driving the observed patterns is limited.

PERMANOVA tables of the full untransformed dataset for *Pocillopora verrucosa* for (i) differences between impacts and (ii) differences between sites, and for *Acropora hemprichii*

for (iii) differences between impacts and (iv) differences between sites (these are not included in the revised manuscript):

Table i: Bacterial microbiome differences across anthropogenic impacts for the coral *Pocillopora verrucosa* after 21 months of reciprocal transplantation. OTU count data untransformed, based on Bray-Curtis distances.

Factors	Df	SumsOfSqs	MeanSqs	F.Model	R2	Pr(>F)
destination_impact	2	0.7151	0.35755	2.49649	0.10193	0.0115
origin_impact	2	0.1973	0.09865	0.68878	0.02812	0.7681
destination_impact:origin_impact	4	0.5179	0.12948	0.90407	0.07382	0.5655
Residuals	39	5.5857	0.14322	0.79613		
Total	47	7.016	1			

Table ii: Bacterial microbiome differences across reef sites for the coral *Pocillopora verrucosa* after 21 months of reciprocal transplantation. OTU count data untransformed, based on Bray-Curtis distances.

Factors	Df	SumsOfSqs	MeanSqs	F.Model	R2	Pr(>F)
destination_site	4	1.0923	0.27308	1.91577	0.15569	0.0258
origin_site	4	0.471	0.11774	0.826	0.06713	0.7007
destination_site:origin_site	15	2.0316	0.13544	0.95017	0.28957	0.583
Residuals	24	3.4211	0.14255	0.48761		
Total	47	7.016	1			

Table iii: Bacterial microbiome differences across anthropogenic impacts for the coral *Acropora hemprichii* after 21 months of reciprocal transplantation. OTU count data untransformed, based on Bray-Curtis distances.

Factors	Df	SumsOfSqs	MeanSqs	F.Model	R2	Pr(>F)
destination_impact	2	2.558	1.27898	5.1225	0.1686	0.0001
origin_impact	2	0.4795	0.23977	0.9603	0.03161	0.5018
destination_impact:origin_impact	4	0.8991	0.22477	0.9002	0.05926	0.6752
Residuals	45	11.2356	0.24968	0.74054		
Total	53	15.1722	1			

Table iv: Bacterial microbiome differences across reef sites for the coral *Acropora hemprichii* after 21 months of reciprocal transplantation. OTU count data untransformed, based on Bray-Curtis distances.

Factors	Df	SumsOfSqs	MeanSqs	F.Model	R2	Pr(>F)
destination_site	4	3.676	0.91899	4.0162	0.24228	0.0001
origin_site	4	1.1762	0.29404	1.285	0.07752	0.0848
destination_site:origin_site	14	3.2265	0.23046	1.0072	0.21266	0.4566
Residuals	31	7.0935	0.22882	0.46754		
Total	53	15.1722	1			

2. Directly prompted by the reviewer's comment, we repeated the analyses excluding Endozoicomonadaceae from the dataset entirely. Again, the large-scale patterns remained the same between this reduced dataset and the full dataset (at an identical subsampling cutoff of 3,101 sequences per sample).

In the revised manuscript, we include the results of this additional analysis in the supplement and in the manuscript text itself:

Materials and Methods: "Given that the microbiome of *P. verrucosa* was dominated by a single bacterial taxon (OTU0001, Endozoicomonadaceae), a third dataset was created that entirely excluded the Endozoicomonadaceae family with the command `remove.lineages()` in `mothur` as detailed above. This dataset was then analyzed with the same pipeline as the full dataset and confirmed the observed differences in microbiome flexibility between the two species (Source Data)."

Results: "After 21 months of reciprocal transplantations between 5 reef sites, the microbiome of *A. hemprichii* differed significantly between all individual sites. By comparison, differences in the microbiome of *P. verrucosa* between sites were far less pronounced, with only fragments transplanted to the most highly impacted site (municipal wastewater outfall - site E) being significantly different from coral fragments at the unimpacted sites A and B. The latter were also significantly different from each other (Table 2). The differential pattern of microbiome restructuring between coral species was also evident when sites were pooled by impacts (Source Data 7). Notably, analyzing the dataset using a 99% OTU similarity cutoff to account for putative differences at a higher phylogenetic resolution confirmed observed patterns between sites and impacts (pooled sites) (Source Data 5, 7). Similarly, excluding Endozoicomonadaceae from the dataset to rule out that patterns were largely driven by dominant association with *Endozoicomonas* also reproduced that patterns are different between both coral species when considering microbiome composition across impacts (pooled sites) (Source Data 6 - 7)."

Results on alpha diversity: "Notably, re-analyses of the dataset at a 99% OTU similarity cutoff recaptured the observed patterns (Figure S9, Source Data 9). However, it should be noted that because the Endozoicomonadaceae are a dominant feature in the *P. verrucosa* microbiome, excluding bacteria of this family changed the evenness and diversity of these microbial communities (Figure S9; Source Data 10)."

Discussion: “The most abundant *Endozoicomonas* OTU in *P. verrucosa* dominated most samples and, in contrast to the loss of Endozoicomnadaceae in *A. hemprichii* at impacted sites, was consistent. Generally, there were no major abundance changes in bacterial taxa between impact levels in *P. verrucosa*, with the exception of the Simkaniaceae family whose role we can only speculate on. Additional analyses excluding the Endozoicomnadaceae family further support the notion of a more stable and less variable microbiome of *P. verrucosa* compared to *A. hemprichii*. Whether dominance of a particular bacterial lineage promotes a generally more stable microbiome (inducing stability of less abundant members), or whether it is simply the indication of an inflexible host-microbial association remains to be determined.”

We are therefore confident that the pattern we report on reflect the underlying biology. The additional analyses put forward in the revised version reinforce the validity of the observed pattern of differential microbiome flexibility between the two coral species.

Second, it is very likely the case that comparing the microbiomes of these two corals is essentially akin to comparing apples and oranges. It is known that *Endozoicomonas* are generally found inside the tissues of the coral animal, and has such should not be expected to respond to environmental stimuli in the same manner as microorganisms residing exterior to the coral animal tissue, or respond on the same temporal scale. Is it particularly meaningful to compare a microbiome known to be dominated by an intracellular monoculture to one where the location of microbial cells is not known, but is probably dominated by cells that reside on the exterior of the coral tissue?

It is true that the tissue location of most coral-associated bacteria is currently unknown. Both coral species in this study however are associated with dominant Endozoicomnadaceae at unimpacted sites. In *A. hemprichii*, this association is not persistent, while in *P. verrucosa*, it is. We still cannot unequivocally rule out that this plays a role, but we previously observed analogous differences in flexibility between coral taxa investigating global patterns of *Endozoicomonas* in *Stylophora pistillata* and *Pocillopora verrucosa* (Neave et al. 2017): here both coral host species harbor highly abundant intracellular *Endozoicomonas*, yet *P. verrucosa* showed a ‘globally conserved’ pattern, i.e. associates with similar *Endozoicomonas* taxa over geographical scales, whereas *S. pistillata* showed a spatially distinct pattern, i.e. associates with distinct *Endozoicomonas* taxa over geographical scales. This supports the argument that microbiome flexibility is different between species and, even if biased to some degree by dominant bacteria and their physical location, cannot be alone and sufficiently explain the differences in microbiome structure.

As an additional argument to put forward: there are few other bacteria that have been shown to reside within coral tissues, such as Actinomycetales and Burkholderiales (Ainsworth et al. 2015). Both of these lineages are also present in our dataset (albeit at lower abundances) in both coral species, thus arguing that differences cannot be exclusively attributed to physical location.

From a sample processing point of view, we have taken the precaution of rinsing the coral fragments with filtered seawater before freezing them, with the aim to reduce externally associated bacteria, biofilms, and contaminants. The large difference between seawater and coral microbiomes we see (Figure S2) supports the interpretation that coral-associated bacteria are host specific, even if they reside externally.

Ainsworth, T., Krause, L., Bridge, T., Torda, G., Raina, J.-B., Zakrzewski, M., . . . Leggat, W. (2015). The coral core microbiome identifies rare bacterial taxa as ubiquitous endosymbionts. *ISME J*, 9(10), 2261-2274

Neave, M. J., Rachmawati, R., Xun, L., Michell, C. T., Bourne, D. G., Apprill, A., & Voolstra, C. R. (2017). Differential specificity between closely related corals and abundant *Endozoicomonas* endosymbionts across global scales. *ISME J*, 11(1), 186-200

Specific points:

Lines 52-54: no experiments were performed to assess whether a change in microbiome is a strategy for coping with environmental change. This is purely conjecture.

Yes, thank you. We have rephrased this:

We suggest that distinct degrees of microbiome flexibility exist, potentially reflecting different holobiont adaptation mechanisms to respond to environmental change.

Lines 218-219: 97% cutoff for OTUs is now considered quite conservative, and possibly masks some interesting dynamics.

Yes, we agree that this cutoff may be considered relatively conservative. To check that we are not indeed missing fine-scale community patterns, we repeated the sequence and alpha and beta statistical analyses using a 99% OTU similarity cutoff (results of both included as Figure S9, Source Data 5, 7, 9).

The results of this additional analysis are comparable to the results using a 97% similarity cutoff. Notably, the main *Endozoicomonadaceae* OTU using 97% similarity was split into 2 OTUs in *Pocillopora verrucosa*. These OTUs were distributed indiscriminately in our samples and did not influence the observed grouping(s) at large, an observation that fits well with the recently published results by Glasl and colleagues (2019) and the recovery of two *Endozoicomonas* genomes that were assembled from metagenomic data of a single sample of *P. verrucosa* (Neave et al. 2017).

Accordingly, we added the following to the revised manuscript:

“For all subsequent analyses, Operational Taxonomic Units (OTUs) were built based on sequence clustering at a 97 % similarity cutoff. Because this cutoff can be considered rather conservative, a comparative analysis of a second dataset using a 99 % similarity cutoff for OTU clustering was also conducted. This analysis yielded similar sample groupings and results (Source Data).”

Glasl, B., Smith, C. E., Bourne, D. G., & Webster, N. S. (2019). Disentangling the effect of host-genotype and environment on the microbiome of the coral *Acropora tenuis*. *PeerJ*, 7, e6377.

Neave, M. J., Michell, C. T., Apprill, A., & Voolstra, C. R. (2017). *Endozoicomonas* genomes reveals functional adaptation and plasticity in bacterial strains symbiotically associated with diverse marine hosts. *Scientific Reports*, 7, 40579.

Lines 274-285: It is interesting that no abundant cyanobacteria were recovered from the seawater samples. Why is that?

The low abundance of cyanobacteria in our seawater dataset is explained by the 16S primers that we used (784F & 1061R, V5-V6). Our primer choice was determined by the complexity of the coral holobiont environment. We used primers that avoid amplification of 16S chloroplast and host mitochondrial sequences and these primers also do not amplify cyanobacterial 16S very well. The primer choice is based on a comparative analysis testing different primer pairs for their specificity to amplify bacteria (over 18S and 16S sequences originating from host and algal symbionts), published in the supplement of Bayer et al. (2013). The results we obtained for the water samples are in line with previous surveys using these primers (e.g. Roder et al. 2016, Jessen et al. 2014).

Bayer, T., Neave, M. J., Alsheikh-Hussain, A., Aranda, M., Yum, L. K., Mincer, T., . . . Voolstra, C. R. (2013). The microbiome of the Red Sea coral *Stylophora pistillata* is dominated by tissue-associated *Endozoicomonas* bacteria. *Applied and Environmental Microbiology*, 79(15), 4759-4762.

Jessen, C., Villa Lizcano, J. F., Bayer, T., Roder, C., Aranda, M., Wild, C., & Voolstra, C. R. (2013). In-situ effects of Eutrophication and Overfishing on Physiology and Bacterial Diversity of the Red Sea Coral *Acropora hemprichii*. *PLoS ONE*, 8(4), e62091

Roder, C., Bayer, T., Aranda, M., Kruse, M., & Voolstra, C. R. (2015). Microbiome structure of the fungid coral *Ctenactis echinata* aligns with environmental differences. *Molecular Ecology*, 24, 3501–3511

Line 337: Replace “3C” with “3B”.

Thank you for picking this up. This has been adjusted.

Reviewer #2 (Remarks to the Author):

The study from Grupsta et al. is investigating microbiome changes in response to anthropogenic impacts in two different coral species. The author cross-transplanted coral fragments from 5 sides for the duration of 21 month. At the end of 21 months microbiomes were analyzed based on 16S rRNA gene analysis. The authors claim that the two coral species show different “microbiome flexibility”. While one coral responded with large changes in the microbiome, the other species was more stable in its microbiome.

We thank the reviewer for the time to review and comment on our manuscript.

I have two major points that should be considered:

1) The authors argue, that the coral P.v. is less flexible in its microbiome compared to the coral A.h. In Figure 2A all analyzed samples from both corals are depicted. If the interpretation is right, that A.h. is more flexible compare to P.v., then the within treatment distances calculated based on the distance matrix should be significantly smaller in A.h. compared to P.v. Just based on Figure 2A, I would not expect that these values are different.

Thank you for this excellent point! The fact that the samples from both coral species seem equally dispersed in the ordination of Figure 2A can probably be explained by the large influence of the water samples that structure this plot (the coral samples are much closer to each other as indicated by the large distance of coral samples to the water samples). We revised this plot to only include the samples from the two coral species. This plot now nicely illustrates the higher dispersion of samples in *Acropora hemprichii* when compared to *Pocillopora verrucosa*. As such, the reviewer’s notion was spot on and we are happy to have revised the manuscript accordingly to clear this up. We revised Figure 2 to highlight the difference in flexibility between the coral species and added supporting figures to the Supplement (Figure S4-S8).

In addition, we provide new results on differences in “multivariate dispersion” and “distance to group centroids” as a measure of microbiome flexibility in the revised manuscript.

Materials and Methods: “To test microbiome flexibility, ‘betadisper’ function was used to calculate multivariate dispersion of samples (Bray-Curtis distances) between coral species and between impacts within species. Homogeneity of multivariate dispersions were tested with ANOVAs, followed by Tukey's Honest Significant Differences post-hoc test where applicable and visualized with boxplots.”

Results: “A total of 4,704 bacterial OTUs were identified for *A. hemprichii* and 3,023 OTUs for *P. verrucosa*. Of these, 1,628 OTUs (27 %) were shared between the two coral species. Bacterial assemblages were highly species-specific (Figure 2a) and significantly differed in their multivariate dispersion between coral species (ANOVA, $F = 16.01$, $p < 0.001$). Overall, microbiomes of *A. hemprichii* samples were more variable than those of *P. verrucosa* as evidenced by significantly higher distances to centroids (*A. hemprichii* mean = 0.55, SD = 0.05; *P. verrucosa* mean = 0.51, SD = 0.06, Figure S4, Source Data 4).”

Discussion: “We found that the *A. hemprichii* microbiome is highly flexible and more variable, whereas the *P. verrucosa* microbiome is relatively stable and less variable in response to changing environmental conditions.”

In addition, I would expect that in the less flexible coral P.v. “origin site” should explain some of the observed microbiome differences. I agree that the changes in A.h. can be better explained by the destination site, as in P.v. But the general microbiome flexibility seem to be as high in P.v. as in A.h.. I encourage the authors to address this difference, by differentiating between general microbiome flexibility, which seem to be similar between both species (based on Figure 2a), and differences, which are explained by destination site.

Point taken. If microbiomes in *P. verrucosa* were very different between impacts after the transplantation, and following the hypothesis that the microbes are inflexible, we should expect to see a larger influence of the sites of origin in *P. verrucosa* than in *A. hemprichii*. This is however what we are seeing: in the current study the microbiome of *P. verrucosa* is largely similar between all sites and impacts. Because of this similarity, statistical analyses do not resolve significant differences between origins.

One could even argue other way around. The microbiome of A.h. responds to destination side more robust compared to P.v.. In P.v. the microbiome accumulates more stochastic differences over time, that are not explained by destination side, and therefore is more flexible.

We hope that the additional analyses, in particular the measure of dispersion, convince the reviewer that the microbiome of *P. verrucosa* remained more stable between transplantations to different impacts and is overall less variable than that of *A. hemprichii*, which showed significant changes between all sites.

2) The authors state in the abstract (line 49) and in the discussion (line 423) that back-transplants were performed to measure the potential of the microbiome to recover. But in the M& M section it is written that only one cross-transplantation experiment was performed. For a back-transplantation experiment, I would expect a second transplantation after the first 21 month. In addition, I miss a corresponding data analysis in the results section.

We apologize and have clarified this part of the experiment in the manuscript. The back-transplantation was part of the original experiment, in which fragments of each colony were also transplanted back to their site of origin. This was done to control for the handling effect of the transplantation. The data are included for these samples and the statistical analyses show that destination impact (i.e. cross-transplantation) significantly influence the microbiome regardless of site and particular impact the samples were transplanted from. Conversely, transplanting back to the site of origin (i.e. back-transplantation) did not produce significant differences, providing evidence that the differences in bacterial community patterns we see are indeed a product of the transplantation to different sites and not a product of the transplantation procedure itself. We clarified this in the revised manuscript as follows:

“Interestingly, upon cross-transplantation to unaffected sites, we found that microbiomes became indistinguishable from back-transplanted controls, suggesting the ability of microbiomes to recover.”

Previous line 423: We have removed this passage from the beginning of the discussion and focus on it in the last paragraph before the conclusions.

Specific points:

Figure 2A: Are water samples taken from all sides? I would expect, that anthropogenic effects should be reflected also in the water microbiome, as also stated by the authors in line 119. But in Figure 2A all samples are highly similar. Which biotic and abiotic factors are known to be different at the different sides? Temperature, salinity, DOC, etc? The contribution of measured environmental data should be added to the analysis in addition to destination side, which is not well-defined.

Thank you for the observation. The water samples are indeed much more similar to each other than host associated samples are to each other (Figure S2). This observation is consistent with our previous studies, in particular Ziegler et al (2016). Indeed, we see only few changes in bacterial communities of seawater between the sites, and this can possibly be explained by three factors: 1) the sampling took place at a single time point at the end of the transplantation experiment, which may not reflect long-term differences at these sites. 2) Generally, the microbial communities in the seawater are better connected than those within the corals, therefore the lack of large differences does not come as a total surprise. 3) Because of the complex environment of the corals (to avoid/minimize host amplification), we have chosen a primer set that may not be ideal in picking up water contaminants (e.g. coliforms) and may not resolve these differences particularly well.

To strengthen this part of the manuscript, we now include more detail and information from the study sites based on previous data and further include new additional data of three environmental variables (nitrate, sedimentation, linear alpha olefin):

“Sites A and B were relatively unimpacted and represent almost pristine control conditions. Both locations were characterized by comparatively low sedimentation loads (Figure S1, Source Data 1), low inorganic nitrate concentrations (Figure S2, Source Data 2), and low levels of total hydrocarbons (THC), measured against a standard of Light Arabian Crude Oil (Figure S3, Source Data 3; see supplementary Materials and Methods for details on measurements). During previous surveys a high stony coral cover and diversity as well as low abundances of soft corals was recorded at these sites (Ziegler et al., 2016). Sites C and D were located along the fringing reef of the heavily developed Jeddah Corniche and represented an intermediate impact level. The area is exposed to chronic turbidity and intermediate sedimentation loads from infilling paired with unauthorized local wastewater outfalls that are estimated to release 99,000 m³ d⁻¹ of untreated wastewater into the nearshore area along 30 km of coastline and lead to elevated nitrate levels (Figure S2, Source Data 2)(Al-Farawati, 2010; Peña-García et al., 2014; Risk et al., 2009), while levels of THC are comparable to the unimpacted sites (Figure S3, Source Data 3). Both sites were characterized by relatively high cover of *Xenia* spp., known to opportunistically invade degraded reefs (Ziegler et al., 2016; Benayahu and Loya, 1985; Tilot et al., 2008). Site E represents the most severe impact level being located within Jeddah Bay, in proximity to the industrial port which generates intermediate levels of oil pollution (Figure S3, Source Data 3). In

addition, site E was less than five km from the three main discharge points of Jeddah's sewage and treatment facilities, which regularly discharge extensive amounts (35,000, 68,000, and 300,000 m³ d⁻¹, respectively) of untreated or only partially treated sewage and lead to intermittent increases in nitrate levels (Figure S2, Source Data 2)(Basaham et al., 2009; El-Rayis and Moammar, 1998; Mudarris and Turki, 2006). Hence, this site is subjected to elevated turbidity and highest sedimentation loads (Figure S1, Source Data 1). Hard coral cover at site E is similar to sites C and D and soft coral cover abundances were intermediate, with lower counts of *Xenia spp.* (Ziegler et al., 2016).”

Figure 2B,C: In Table 2 it is shown that also the comparison of destination sides A vs.B are significantly affecting microbiome structure. Therefore, I would suggest coloring all destination sides with unique colors throughout the manuscript.

We used two different strategies to distinguish ‘impacts’ and ‘sites within impacts’ in the manuscript. The initial plan was to use replicated sites for each impact. However, due to loss of site 6 (most likely due to entanglement in fishing gear), we were left with deciding how to best address this issue. To present the data in the most genuine way, we decided to keep both levels. Impacts are denoted as colors and sites with symbols. We hope that this approach facilitates easy-to-grasp illustration of the data structure. In the revised manuscript, we added different shades of the respective impact colors to further highlight sites (e.g. light and dark blue for the unimpacted sites A and B respectively). We also added different versions of these ordination plots illustrating destination & origin sites and origin sites only as supplementary figures (Fig S5-S8).

Line 354: Based on Figure 4 the authors argue that there is a qualitative difference in the abundance changes between A.h. and P.v.. In my opinion, bar charts are not the adequate method to illustrate that. I encourage the authors to show the LEfSe analysis e.g. by using LefSe cladograms. The qualitative differences in the responsiveness between A.h. and P.v. do not get evident in the bar charts.

Thank you for this suggestion. We have previously attempted to publish similar data in the form of a LEfSE cladogram (Ziegler et al. 2017), and none of the previous reviewers found this way of illustrating the pertinent changes to be very intuitive. We therefore prefer not to use cladograms for this type of data (also, because abundances are not easily depicted in cladograms).

The reviewer picked up on an important point, as the results of the LEfSe analyses support our main results quite well. Therefore, we decided to add the results from the LEfSe analyses as a heatmap (new Figure 4) in the manuscript. This plot highlights the different scales at which bacterial taxa were found to be significantly different between the two coral species (60 taxa in *Acropora*, 5 taxa in *Pocillopora*).

Ziegler, M., Seneca, F. O., Yum, L. K., Palumbi, S. R., Voolstra, C. R. Bacterial community dynamics are linked to patterns of coral heat tolerance. *Nat Commun* 8, 14213 (2017)

Reviewer #3 (Remarks to the Author):

The paper by Grupstra et al aiming to explore microbiome flexibility in corals is certainly of interest and an exciting experiment with lots of promise. However, as its currently written I do not see it being published in this journal. That said I would be more than happy to see a revised version answering or exploring some of the issues I identify below.

We thank the reviewer for their time and evaluation of our manuscript. We have addressed all of the comments individually, some of which arose due to misunderstandings of our experimental procedures. We apologize for this. We have rewritten and clarified the respective passages as detailed below.

Firstly, I think the paper is possibly over stating its novelty. Two similar (but of course less detailed) studies were published a while back by Garren et al. in 2008 and 2009

Garren, M., Raymundo, L., Guest, J., Harvell, C. D., & Azam, F. (2009). Resilience of coral-associated bacterial communities exposed to fish farm effluent. *PLoS One*, 4(10), e7319.

Garren, M., Smriga, S., & Azam, F. (2008). Gradients of coastal fish farm effluents and their effect on coral reef microbes. *Environmental microbiology*, 10(9), 2299-2312.

Which may be worth exploring. I should note at this point just cause I highlight a paper does not mean you have to cite it, just please read it and see if it fits with what you are saying.

Thank you for pointing out these studies that we had previously missed. This was entirely unintended and we are aware that it becomes harder to keep track of all the good literature published, which of course doesn't relieve authors to conscientiously query the existing literature. To remedy this and be more inclusive of the literature, we have now included this aspect dutifully in the revision:

“[...] In addition, some studies have found seasonal fluctuations in coral-associated microbiomes (Koren and Rosenberg, 2006; Li et al., 2014; Roder et al., 2015) and tide-related shifts on much shorter time scales (Sweet et al., 2017), while other corals maintain temporally stable microbiomes (Chu and Vollmer, 2016). Unidirectional transplantation experiments of the coral species *Acropora muricata* (Casey et al., 2015) and *Porites cylindrica* (Garren et al., 2009) from a pristine site to impacted or modified sites illustrate that microbiomes change under adverse environmental conditions. Importantly, Ziegler et al. (2017) showed that microbiomes of heat tolerant *Acropora hyacinthus* can be acquired by heat sensitive corals upon environmental transplantation over the course of 17 months. Notably, these corals exhibit increased thermotolerance in a subsequent heat stress experiment, harboring a more robust and stable microbiome.

At present, it is unclear whether the potential for microbiome restructuring is a conserved trait across coral species, or whether species-specific differences exist. For instance, the coral *Pocillopora verrucosa* shows a globally conserved association with its main bacterial symbiont *Endozoicomonas* (Neave et al., 2017) that remains unchanged even under conditions of bleaching and mortality (Pogoreutz et al., 2018). *P. verrucosa* further maintained a stable Symbiodiniaceae community during a cross-transplantation experiment over depth (Ziegler et al., 2014) and between seasons and reefs, while *Porites lutea* sampled under the same conditions had a highly flexible Symbiodiniaceae community (Ziegler et al., 2015). Therefore, it appears that

the ability of corals to associate with distinct microbial associates may depend on geographical variation, environmental setting, and coral host species.”

As the reviewer notes, the above-mentioned studies are different with regard to scale, scope, and methods. Importantly, these papers do not systematically elucidate species-specific microbial flexibility patterns *per se*, but rather make a case that microbial association is variable under adverse environmental conditions. As such, these studies lack the fully-crossed transplant design, and we would argue that they do not compromise the novelty of our findings.

The studies by Garren and colleagues do a great job at quantifying the environmental impact of the fish farms, but are rather limited in methodological scope with regard to coral microbiomes, specifically:

The study Garren et al. 2008 took samples from coral species and fish feces along an environmental gradient analyzing microbial profiles using DGGE and clone libraries, but was constrained by the absence of live coral throughout the transect (3 sampling sites where corals were sampled, site B: 4 samples of *Porites* sp., site C: 1 sample of *Porites* sp. & 1 sample of *Agaricia*, site D: 1 sample of *Fungia* sp.).

The follow-up study by Garren et al. 2009 used four colonies of the coral *Porites cylindrica* and transplanted fragments from these colonies from one site (the unimpacted control) to several other sites to investigate the effect of fish farm effluent on the microbiome using DGGE and clone libraries.

As such, these studies can be considered pioneer studies and set up the framework for a study like ours to specifically test/address the hypothesis whether coral species do bear differing levels of structural flexibility with regard to microbiome composition. Accordingly, they should be cited and mentioned in our work, which we amended in the revised version.

Furthermore, a couple of studies by Sweet et al. show both short term mechanical disruption and recovery of the microbiome and more natural variation over tidal stress – one which you rightly cite but could explore in more detail maybe. Also from the same group they highlight how the microbiome varies with age, this might be interesting to at least note and discuss later and its implications for your study and the age of your corals sampled.

Sweet, M. J., Croquer, A., & Bythell, J. C. (2011). Dynamics of bacterial community development in the reef coral *Acropora muricata* following experimental antibiotic treatment. *Coral Reefs*, 30(4), 1121.

Sweet et al (2013). Changes in microbial diversity associated with two coral species recovering from a stressed state in a public aquarium system. *Journal of Zoo and Aquarium Research*, 1(2), 52.

Williams, A. D., Brown, B. E., Putschim, L., & Sweet, M. J. (2015). Age-related shifts in bacterial diversity in a reef coral. *PLoS One*, 10(12), e0144902.

Thank you for these suggestions. We also think that it is important to disentangle time, colony age, and handling from treatment effects. In our experimental design we use back-transplanted colonies to control for these factors. We added more information on this to the revised manuscript accordingly:

“This design allowed fragments from each colony to be assessed at each location, including one fragment per colony that was back-transplanted to its respective site of origin. Of note, the back-transplanted fragments (to their site of origin) act as experimental and handling ‘controls’ in that any microbiome changes would arguably be a result of the transplant procedure. These colonies also aid in correcting for possible changes of the microbiome that may be related to time or age of the fragments, in that fragments from the same colonies (with the same age, sharing the same life history) were investigated at all sites. (Williams et al., 2015)”

We further include background to clearly distinguish short-term ‘disruption’ of the microbiome from long-term or chronic impacts and the response to these upon reciprocal transplantation.

“The use of a reciprocal transplant design between reef sites subjected to different levels of anthropogenic impact allowed us to assess whether environmental differences align with distinct microbiomes and whether the ability to adapt microbiomes differs between coral species. As coral microbiomes have previously been shown to recover from stress events, such as bleaching (Bourne et al., 2007) or disease (Sweet et al., 2013), we were also interested to elucidate whether coral microbiomes can recover from chronic pollution, i.e. return to a state that resembles conspecific microbiomes at unaffected sites upon transplantation of coral fragments from affected to pristine sites.”

Introduction

I think the intro could be a little more topical and inclusive of the current literature, some highlighted above. Furthermore you should rename *Symbiodinium* in accordance with the new classification which at least one of the authors of this publication was involved in ;)

Thank you for this suggestion, we rewrote large parts of the introduction to include a broader selection of the current literature (detailed above) and have of course renamed *Symbiodinium* according to the new nomenclature, which was published during the review process of this manuscript.

Referencing throughout also needs to be addressed for example L73 but quite a few other examples and some mistakes in the ref list.

Thank you for picking this up, we revised the in-line references and reference list.

L85 enjoy is a little to human feelingy – maybe exhibit or show

Thank you for this suggestion, we rephrased the section:

“[...] these corals exhibit increased thermotolerance in a subsequent heat stress experiment [...]”

Finally a third of your intro is put over to what you did – usually this is only a small part highlighting your main aim and I feel you make more use of the literature to tell a stronger and more compelling story about why you need to do what you have done. For example you explore what we know about the corals structuring of the microbiome, see Sweet et al 2011 and a more recent paper by Pollock which I believe just came out <https://www.nature.com/articles/s41467-018-07275-x>.

Sweet, M. J., A. Croquer, and J. C. Bythell. "Bacterial assemblages differ between compartments within the coral holobiont." *Coral Reefs* 30.1 (2011): 39-52.

The reviewer is correct. We have shortened the last part of the introduction on what we did and added more references to highlight the context of our experimental questions:

“To assess potential differences in flexibility of microbial association across coral species, we conducted a long-term large-scale reciprocal transplantation experiment using the coral species *A. hemprichii* and *P. verrucosa*. Based on previous studies, these coral genera were suspected to differ in the flexibility of their association with different microbial communities across environmental gradients (Jessen et al., 2013; Neave et al., 2017 ; Pogoreutz et al., 2018; Ziegler et al., 2016). ~~As part of a wider study of the effects of anthropogenic impacts on Saudi Arabian coral reefs, we conducted back and cross transplantation of coral colony fragments from both species between five sites with differing levels of shore based pollution near Jeddah, the largest city in the Red Sea. Subsequently we performed 16S rRNA gene amplicon sequencing to assess the impact on microbial community structure.~~ The use of a reciprocal transplant design between reef sites subjected to different levels of anthropogenic impact allowed us to assess whether environmental differences align with distinct microbiomes and whether the ability to adapt microbiomes differs between coral species. As coral microbiomes have previously been shown to recover from stress events, such as bleaching (Bourne et al., 2007) or disease (Sweet et al., 2013), we were also interested to elucidate whether coral microbiomes can recover from chronic pollution, i.e. return to a state that resembles conspecific microbiomes at unaffected sites upon transplantation of coral fragments from affected to pristine sites.”

In the Discussion we included the aspect raised by the recently published paper by Pollock et al. as pointed out by the reviewer:

“Potentially relevant here is that the coral phylogeny is characterized by a deep phylogenetic split (between the ‘complex’ and ‘robust’ clades) that dates back > 245 mya, resulting in substantial genomic divergence with consequences for host-microbe pairings that may be driven by phyllosymbiosis (Pollock et al., 2018)”

Materials and Methods

My first question was where is F but then you explain it got wiped out, I might suggest therefore to remove it completely as you give it no further consideration afterwards inc in figure 1

The reviewer picked up on a point that we were debating before submission as we feel that it is important to mention at least that site F existed (before being wiped out) to explain the unbalanced design with regard to ‘duplicated’ impacts over sites. This is why the second sentence of the Materials and Methods states that the 6th site was destroyed and that site has also not been included in Figure 1. As it stands, we would like to keep this information, but are happy to remove it, should the reviewer request it.

You mention differences in benthic community, seawater microbes etc, are these highlighted in Zieglers paper? If so say so or maybe make a supplementary table

We added a more detailed site description to the Materials and Methods and also include additional new data quantifying the anthropogenic impacts (Supplementary Data):

“Sites A and B were relatively unimpacted and represent almost pristine control conditions. Both locations were characterized by comparatively low sedimentation loads (Figure S1, Source Data 1), low inorganic nitrate concentrations (Figure S2, Source Data 2), and low levels of total hydrocarbons (THC), measured against a standard of Light Arabian Crude Oil (Figure S3, Source Data 3; see supplementary Materials and Methods for details on measurements). During previous surveys a high stony coral cover and diversity as well as low abundances of soft corals was recorded at these sites (Ziegler et al., 2016). Sites C and D were located along the fringing reef of the heavily developed Jeddah Corniche and represented an intermediate impact level. The area is exposed to chronic turbidity and intermediate sedimentation loads from infilling paired with unauthorized local wastewater outfalls that are estimated to release 99,000 m³ d⁻¹ of untreated wastewater into the nearshore area along 30 km of coastline and lead to elevated nitrate levels (Figure S2, Source Data 2)(Al-Farawati, 2010; Peña-García et al., 2014; Risk et al., 2009), while levels of THC are comparable to the unimpacted sites (Figure S3, Source Data 3). Both sites were characterized by relatively high cover of *Xenia* spp., known to opportunistically invade degraded reefs (Ziegler et al., 2016; Benayahu and Loya, 1985; Tilot et al., 2008). Site E represents the most severe impact level being located within Jeddah Bay, in proximity to the industrial port which generates intermediate levels of oil pollution (Figure S3, Source Data 3). In addition, site E was less than five km from the three main discharge points of Jeddah’s sewage and treatment facilities, which regularly discharge extensive amounts (35,000, 68,000, and 300,000 m³ d⁻¹, respectively) of untreated or only partially treated sewage and lead to intermittent increases in nitrate levels (Figure S2, Source Data 2)(Basaham et al., 2009; El-Rayis and Moammar, 1998; Mudarris and Turki, 2006). Hence, this site is subjected to elevated turbidity and highest sedimentation loads (Figure S1, Source Data 1). Hard coral cover at site E is similar to sites C and D and soft coral cover abundances were intermediate, with lower counts of *Xenia* spp. (Ziegler et al., 2016).”

L156 swab approx. and every around

Edited accordingly.

Furthermore, did you also take a sample from the ‘parental’ colony before taking the colony and fragging it –you could have then compared your back transplanted sample to this to explore any observable change – which I imagine there very well may have been

We expect the microbiome of the corals to (naturally) vary to a certain extent over the almost two years of the transplantation (due to seasonal and age changes). Thus, any changes between fragments and mother colony could be due to time and not treatment effects. To control for handling and treatment, we used the back-transplantation of fragments to their site of origin, which resembled their cross-transplanted conspecifics at the same site. Further, comparing microbial community structures determined in the current study with data from the Ziegler et al

(2016) study (where we directly collected and processed corals from the different sites), shows that the microbial communities were similar, and thus, probably little affected by the handling and transplantation treatment.

Ziegler, M., Roik, A., Porter, A., Zubier, K., Mudarris, M. S., Ormond, R., & Voolstra, C. R. (2016). Coral microbial community dynamics in response to anthropogenic impacts near a major city in the central Red Sea. *Marine Pollution Bulletin*, 105(2), 629-640

L206 some worry about excluding chloroplasts – I'm in two minds but might be worth thinking about this

The primer set we used was chosen (amongst other factors) to avoid amplification of chloroplastic 16S (to avoid amplification of Symbiodiniaceae chloroplasts). Out of the 2.5 million sequences past QC, 718 sequences were annotated as chloroplasts and removed accordingly (average of 5 sequences per sample). In Bayer et al. (2013), we tested primers for their ability to amplify bacterial 16S in corals and that avoid considerable amplification of host mitochondria and dinoflagellate chloroplast 16S in order to maximize sequence data return.

Bayer, T., Neave, M.J., Alsheikh-Hussain, A., Aranda, M., Yum, L.K., Mincer, T., Hughen, K., Apprill, A., Voolstra, C.R., 2013. The microbiome of the Red Sea coral *Stylophora pistillata* is dominated by tissue-associated *Endozoicomonas* bacteria. *Appl. Environ. Microbiol.* 79, 4759-4762.

L2015 – I got a little confused why you started classifying with Silva then moved to greengenes?

The reviewer noticed this correctly. We first aligned our sequenced against SILVA database for the chimera and QC steps and then we use Greengenes database as a reference for annotation of the sequences. In our analysis we followed the mothur MiSeq SOP, and the answer to the reviewer's question can be found on the mothur wiki webpage: "Because of the poor alignment quality in the variable regions [in Greengenes] we strongly discourage people from using it for their "real" analysis. One side effect of this is that chimera.slayer detects fewer real chimeras when using greengenes-aligned sequences compared to SILVA-aligned sequences. "

After using SILVA for QC, annotation & filtering of sequences with Greengenes is then acceptable. The advice is to use the reference database of choice that has best coverage of whatever type of dataset one analyzes. Although the alignment in Greengenes is not as long (only 7,682 columns), it has the largest collection of sequences: 202,421 bacterial and archaeal sequences, which in our experience also contains a higher number of coral-associated (or close to coral-associated) taxa (e.g. see Sunagawa et al. 2009). SILVAs alignment is more accurate and longer (50,000 columns), but it contains fewer reference sequences (168,111 bacteria, 4,337 archaea, and 18,213 Eukarya sequences). In short, we filter conservatively, but annotate more liberal to gain maximum insight.

Sunagawa, S., DeSantis, T. Z., Piceno, Y. M., Brodie, E. L., DeSalvo, M. K., Voolstra, C. R., . . . Medina, M. (2009). Bacterial diversity and White Plague Disease-associated community changes in the Caribbean coral *Montastraea faveolata*. *ISME J*, 3(5), 512-521

Great that you explore core or stable microbiomes but there are plenty of issues with this and you have possibly fallen into one of the traps highlighted by Sweet and Bulling 2017

Sweet, M. J., & Bulling, M. T. (2017). On the importance of the microbiome and pathobiome in coral health and disease. *Frontiers in Marine Science*, 4, 9.

It might be worth running your analysis with 90% and 100% - in my opinion 75% is a bit of a random number and suggests you either have a lack of confidence in your sample collection and prep or of your sequencing if you use any cut off to describe a core microbe. But this is obviously just an opinion.

As the reviewer correctly points out, exploring a coral's core microbiome is a complex topic, enough to fill multiple research papers and reviews. We argue that a cutoff of 75% (used here) is just as indiscriminate as a cutoff of 90 %, simply because we lack a universal consensus on this topic. Yet, 75 % is comparatively conservative compared to other much regarded studies on the topic (e.g. Ainsworth et al. 2015 used 30%). Most importantly, however, we applied a condition that is also alluded to in the mentioned paper by Sweet and Bulling: we require that members of the core microbiome are being present in 75 % of samples at each site, which ensures that these taxa are evenly distributed across all samples. In the revision, we highlight this:

“Such putative core microbiome members were determined for each species separately by querying all bacterial OTUs that were present in ≥ 75 % of samples at each site, ensuring even distribution of these core taxa across the sample set (Sweet and Bulling, 2017).”

We further provide a full list of this proposed core microbiome in the supplement (Table S3), which details the actual presence and abundance across samples per site. This list of course also contains all taxa that are present in 90 or 100 % of samples (and therefore has the advantage of being more insightful and inclusive) than at a higher cutoff.

Ainsworth, T., Krause, L., Bridge, T., Torda, G., Raina, J.-B., Zakrzewski, M., ... Leggat, W. (2015). The coral core microbiome identifies rare bacterial taxa as ubiquitous endosymbionts. *ISME J*, 9(10), 2261-2274

Sweet, M. J., & Bulling, M. T. (2017). On the importance of the microbiome and pathobiome in coral health and disease. *Frontiers in Marine Science*, 4, 9.

Results

L146-155 I worry about your level of replication – I started to have concerns here and as I read more and more they amplified see later comments. if I understand correctly you only have 5 frags and 5 sites. So that means one frag per site and 3 in total as you had three different colonies. My point is with the level of variation seen in the microbiome of corals in close proximity to each other, even in a healthy state this can compound your results and I think if you presented your data in a way which I will suggest later this would do exactly that.

The experimental design is based on 30 coral colonies, 15 colonies per species (36 and 18, respectively, if we include the lost site F). Each of these colonies was split into 6 fragments for a fully crossed design that was based on 216 reciprocally transplanted coral fragments over roughly 60 km of coastline. These numbers reflect what is logistically do-able in the field, and in our assessment provide a reasonable level of replication (in particular, because we have included back-transplants in the experimental design).

Be it as it may, we thank the reviewer for this comment, as this was another point of debate amongst our team. We feel that it is more ‘honest/transparent’ to show the site-by-site results and

differences based on the PERMANOVA, but we also see the problem of replication the reviewer points out. To address this, we decided to add the impact-by-impact statistical results (Source Data 7) to the revised manuscript, in addition to the site-by-site analyses. In short, these analyses significantly increase the number of samples for each tested impact and confirm the site-based results. We added a corresponding explanation and statement to the revised manuscript.

Results: “The differential pattern of microbiome restructuring between coral species was also evident when sites were pooled by impacts (Source Data 7). Notably, analyzing the dataset using a 99% OTU similarity cutoff to account for putative differences at a higher phylogenetic resolution confirmed observed patterns across sites and impacts (pooled sites) (Source Data 5, 7). Similarly, excluding Endozoicomonadaceae from the dataset to rule out that patterns were largely driven by dominant association with *Endozoicomonas* also reproduced that patterns are different between both coral species when considering microbiome composition across sites and impacts (pooled sites) (Source Data 6 - 7).”

Table 1 was exactly why I was worried about your sample strategy and the loss of fragments – therefore I feel that your conclusions are being stretched from relatively thin data

As detailed above, in addition to the ‘over sites’ analysis, we added an ‘over impacts’ analysis, which dramatically increased the number of replicates, and confirmed the patterns we found. We further amended Table 1 to also include the sample numbers per origin site and origin impact (to the right of the table), which should alleviate some concerns regarding sample replication.

L297 I’m not at all surprised by this as arguably a huge percentage of the corals ‘microbiome’ are transients – see above review and likely driven by the environment, whilst the core, possibly truly functioning microbes will not be and this is where the real interest and shifts and changes lie in my opinion

The group of Prof. Tracy Ainsworth has promoted a concept that represents somewhat a middle ground between the reviewer’s point that only the truly ubiquitous taxa are functionally relevant and the alternative notion that all taxa may potentially play a role. Hernandez-Agreda et al. (2016) suggest to divide coral-associated bacteria into three groups: the ubiquitous core microbiome, transient taxa, and environmentally explicit core microbes filling specific functional niches in a given environment. In a way, this concept is also the essence of the promise of coral probiotics (Rosado et al. 2018).

In our study, the results from the LEfSe and the indicpecies analyses address exactly this point. We identified bacterial taxa that are environmentally explicit. Interestingly, the microbiome of *A. hemprichii* seems to be more amenable to such environmentally explicit taxa, while the microbiome of *P. verrucosa* is not (as highlighted in the high and low number of explicit taxa, respectively). Notably, bacterial taxa don’t need to be tightly or constantly associated to be functional. Rather the current notion supports that the holobiont (or metaorganism for that matter) is not static, but fluidic (pending environmental, age, sex, etc. differences), and different microbes can play different roles under different settings (reviewed in Jaspers et al, 2019).

Taken together, the reviewer points out an important aspect that we elaborate on more extensively in the Discussion of the revised manuscript:

“Nonetheless, other bacterial families that were more abundant in *A. hemprichii* at the impacted sites (Erythrobacteraceae, Comamonadaceae, Oxalobacteraceae, Moraxellaceae) have also been isolated from healthy corals, which illustrates that shifts in microbial abundances do not exclusively reflect a pathobiome, but rather an environmentally selected, more beneficial microbiome (Barott et al., 2011; Glasl et al., 2017; Kvennefors et al., 2010; Nelson et al., 2013; Sunagawa et al., 2010). Indeed, the notion of an environmentally explicit set of bacterial taxa that fill specific functional niches through changes in the microbiome of *A. hemprichii* (Hernandez-agreda et al., 2016), is supported by the high number of specific taxa found under different impacts in our LEfSe and indicpecies analyses. In contrast, the number of such environmentally explicit taxa in *P. verrucosa* was an order of magnitude lower (5 and 7 taxa in *P. verrucosa* vs. 60 and 62 taxa in *A. hemprichii* for the indicpecies and LEfSe analyses, respectively). This is also in line with the current notion that holobiont composition (or metaorganism structure for that matter) is not static, but rather dependent on age, development, sex, and environment, among other factors. As such, consistent or close association of bacteria with their animal hosts is not a *sensu stricto* criterion for functional relevance (Jaspers et al 2019).”

Hernandez-Agreda, A., Leggat, W., Bongaerts, P., & Ainsworth, T. D. (2016). The Microbial Signature Provides Insight into the Mechanistic Basis of Coral Success across Reef Habitats. *mBio*, 7(4), e00560-00516

Jaspers, C. *et al.* Resolving structure and function of metaorganisms through a holistic framework combining reductionist and integrative approaches (2019). *Zoology*, online early, doi.org/10.1016/j.zool.2019.02.007.

Furthermore, in Fig 2 how do we know the origin of the samples? I think it would be really useful to present these results i.e. the original coral microbiome profile, the back transplanted one, and then the change that frag goes through when transported to site B, C,D etc

For the purpose of the figures in the manuscript, we marked samples according to their site of destination to support the statistically significant differences of this aspect. We examined several ways of displaying the data and found that to use a combination of origin & destination sites becomes very convoluted and difficult to grasp visually. We therefore added respective ordination plots showing samples by origin and origin & destination in the supplement (Fig. S5-S8).

Table 2, if I understand it means some of your conclusions may well be slightly off mark. Are the horizontal A,B,C etc the origin site? If so please highlight this. So A frags showed no change when transplanted to sites, B, C, D or E if I read this table this way. So in the discussion when you state ‘corals microbiomes recover to their original state when local sources of pollution are removed’ L424. If true then this is what Garren found in the earlier papers so please site, but my interpretation of the table you present suggests something different. .e. if this was true I would expect corals from E transferred to A or B to show no significance in their microbiome profiles but in your table 2 they are the opposite. This got me very confused, so I stopped reading the discussion at line 428 and would recommend a revision of the work and resubmission depending on what the other reviewers and editor suggest.

We apologize if Table 2 was not clear. We have added additional information to clarify the layout of the table. As only the differences between destination sites were significant (main effects test, listed in the left half of the table), the right part of the table shows the outcomes of

the pairwise tests for destination sites (corals transplanted to A compared to those transplanted to other sites irrespective of their origin, etc.).

The last paragraph of the Discussion before the Conclusions considers the aspect of ‘microbiome recovery’, in which we mention the examples given by the reviewer as well as studies from other groups that investigate this topic. As Garren et al. did not test ‘recovery’ of the microbiomes to pristine conditions (they only transplanted from pristine to impacted), we did not include this reference here. But it is included elsewhere in the manuscript (see above explanations).

“A final point arising from results of cross- and back-transplantation is that the microbiomes of both coral species were similar to their local conspecifics after cross-transplantation to unimpacted sites, a finding which may hold the promise of microbiome ‘recovery’. In other words, corals transplanted from impacted to control sites did not, after 21 months, continue to share similarities with sibling colonies that remained at the impacted sites. These findings suggest that stress-induced microbiome alterations may be reverted upon removal of chronic and long-term stressors, similar to the recovery observed after coral bleaching (Bourne et al., 2007) or disease (Sweet et al., 2013). Following the notion that coral microbiomes contribute to coral health, our results indicate that reducing and removing sources of pollution and sedimentation may result in the reversal of microbiomes. Hence, anthropogenic pollution may not irreversibly disrupt microbiomes in supporting coral health as has been suggested (Krediet et al., 2013; Mao-Jones et al., 2010; Rosenberg et al., 2007). Our results are in line with recent studies, which report that increases in coral disease caused by experimental nutrient enrichment were reversed six to ten months after termination of the experimental treatment (Vega Thurber et al., 2014). Taken together, our data create an additional incentive to reduce sources of anthropogenic pollution and sedimentation close to coral reefs, even if the corals on the target reefs already appear stressed and in poor condition.”

That said the results from lines 321 were certainly more interesting than the previous sections (again in my opinion) – I would however err on the side of caution when identifying bacterium to species using a max of 277 bp fragments – I obvs can’t check these but where there really no their blast matches close to these species?

Yes, this is a good point. We have removed the species labels, these are output automatically by the pipeline, but were not meant to be included.

Finally would it not be interesting in Figure 4 to keep A, B, C and D separate and this would allow within ‘treatment’ ‘ecosystem state’ visual analysis to be done by the reader and/or more reliable cross analysis with site E which you could not combine with F for obvious reasons

The visualization separately for origin and destination sites (as opposed to the current figure per impacts, i.e. paired sites) would increase the number of bars four-fold. We think that this would make the visual interpretation of the data harder and not easier. Figure 4 is included in the manuscript to support the statistically significant result for destination impacts. Additional figures showing the full sample-based data are included in the revised manuscript (Supplement Fig. S3).

REVIEWERS' COMMENTS:

Reviewer #2 (Remarks to the Author):

Thanks for the detailed response. I have no further comments.
Sebastian Fraune

Reviewer #3 (Remarks to the Author):

The authors have take on the suggestions from the previous review and that of the other reviewer and in my opinion this has improved the manuscript - it reads very well and I have no suggestions for further edits.

There are a few sentences which could be shortened as some carry on a little and in one part of the discussion there are a few sentences which are underlined for some reason but i'm sure this can/would come out in the type setting/editing stage

its a lovely paper, advances the field of coral microbiome research and a pleasure to read.

regards

NCOMMS-18-29154A

Response to reviewer comments

A point-by-point response is provided below with our responses in blue font.

Reviewer #2 (Remarks to the Author):

*Thanks for the detailed response. I have no further comments. Sebastian Fraune
We thank the reviewer for the positive assessment of our revision.

Reviewer #3 (Remarks to the Author):

The authors have take on the suggestions from the previous review and that of the other reviewer and in my opinion this has improved the manuscript - it reads very well and I have no suggestions for further edits.

There are a few sentences which could be shortened as some carry on a little and in one part of the discussion there are a few sentences which are underlined for some reason but i'm sure this can/would come out in the type setting/editing stage

its a lovely paper, advances the field of coral microbiome research and a pleasure to read.

regards

We thank the reviewer for the kind words and positive assessment of our revision. The underlined sentences in the discussion were indeed a formatting error that was not seen due to the (underlined) track-changes in the document. We have followed the reviewer's advice to shorten long sentences.